# Cloud4D: Estimating Cloud Properties at a High Spatial and Temporal Resolution

**Jacob Lin**
Department of Computer Science
University of Oxford
jacob.lin@cs.ox.ac.uk

**Edward Gryspeerdt**
Department of Physics
Imperial College London
e.gryspeerdt@imperial.ac.uk

**Ronald Clark**
Department of Computer Science
University of Oxford
ronald.clark@cs.ox.ac.uk

## Abstract

There has been great progress in improving numerical weather prediction and climate models using machine learning. However, most global models act at a kilometer-scale, making it challenging to model individual clouds and factors such as extreme precipitation, wind gusts, turbulence, and surface irradiance. Therefore, there is a need to move towards higher-resolution models, which in turn require high-resolution real-world observations that current instruments struggle to obtain. We present Cloud4D, the first learning-based framework that reconstructs a physically consistent, four–dimensional cloud state using only synchronized ground-based cameras. Leveraging a homography-guided 2D-to-3D transformer, Cloud4D infers the full 3D distribution of liquid water content at 25 m spatial and 5 s temporal resolution. By tracking the 3D liquid water content retrievals over time, Cloud4D additionally estimates horizontal wind vectors. Across a two-month deployment comprising six skyward cameras, our system delivers an order-of-magnitude improvement in space-time resolution relative to state-of-the-art satellite measurements, while retaining single-digit relative error ($< 10\%$) against collocated radar measurements. Code and data are available on our project page https://cloud4d.jacob-lin.com/.

## 1 Introduction

Accurately estimating the state of Earth's atmosphere is essential to all aspects of society, from protecting food supplies to routing flights and optimizing renewable power generation. Therefore, there has been much interest in creating machine learning weather and climate prediction systems able to accurately model the evolution of the atmosphere. However, many processes in the atmosphere are too detailed for current models to simulate directly. For example, shallow cumulus clouds generally span less than a kilometer and cover as much as 40% of the Earth's surface (Tselioudis et al., 2021) and they play a central role in controlling the Earth's temperature. As cumulus clouds are small and short-lived, current weather and climate simulators can not resolve them directly. Instead, hand-crafted "parameterizations" are used to approximate the *aggregate* effect of cloud properties on variables such as temperature, moisture, and winds on a much coarser grid. These approximations are a major source of error in both day-to-day forecasts and long-range climate projections. Modern machine-learning systems such as GraphCast (Lam et al., 2023), Pangu-Weather (Bi et al., 2023),

39th Conference on Neural Information Processing Systems (NeurIPS 2025).

and Aardvark (Allen et al., 2025) inherit this problem: they train on "reanalysis" data that comes from the same physics-based models and therefore bake in the same biases.

Improved models are needed, but this is hampered by the difficulty of observing detailed atmospheric phenomena such as shallow cumulus clouds. To develop better models and assess their impact on aggregate environmental variables, we need detailed observations of cloud properties such as the size, spacing, and water content of individual clouds over their whole lifetime (Geerts et al., 2018). However, detailed cloud properties are not easily measured by current observing systems. High-resolution satellite imagery has a revisit time of several days, and while scanning radar and in situ aircraft measurements provide data of high detail, they are typically of a small part of the cloud, lacking the larger context.

In this paper, we introduce Cloud4D, the first learning-based system that reconstructs a physically consistent, four-dimensional cloud state solely from synchronized ground-based cameras. Our method is based on a homography-guided 2D-to-3D transformer architecture that estimates the full spatial distribution of liquid water content and horizontal wind vectors at 25 m resolution and 5 s rate. A two-month real-world deployment with six cameras demonstrates that Cloud4D achieves an order-of-magnitude improvement in space-time resolution over state-of-the-art satellite products while maintaining < 10% relative error against collocated radar retrievals.

Our contributions are threefold:

1. We propose the first method to jointly estimate cloud physical properties (liquid water content, height, thickness) at a high spatial and temporal resolution from multi-view images.

2. To accomplish this, we propose a homography-guided 2D-3D transformer model that ingests images and accurately predicts 3D cloud properties on a high-resolution grid.

3. We demonstrate, on a two-month deployment with six cameras, that our method delivers an order-of-magnitude improvement in temporal resolution over space-borne products while achieving $< 10\%$ relative error against collocated radar retrievals.

## 2   Problem Background

State-of-the-art weather prediction systems, ranging from traditional dynamical models to recent neural-network approaches such as GraphCast (Lam et al., 2023), evolve a core set of variables on a three-dimensional grid. Typical variables include air temperature, pressure, the horizontal wind components, humidity, and average droplet amounts such as cloud water, rain, and snow. However, the models resolve clouds only implicitly and at much coarser resolutions than the scales on which individual cloud elements form and evolve. Our work focuses on estimating these quantities at a much higher resolution using ground-based cameras that can be used as a substitute for radar to help validate and improve weather models.

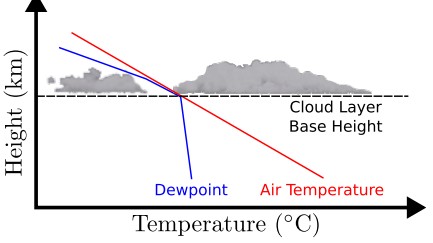

Figure 1: As a result of the cloud formation process, clouds are spatially structured into layers.

The main physical cloud property that we are interested in is the amount of water they contain, as it is a key driver affecting weather and climate. For cumulus clouds, this is characterized by the liquid water content (LWC) ($kgm^{-3}$) distribution, and is a common variable in weather and climate models. As the vertical structure of water in the atmosphere is especially important, the summation of the LWC along a height column is often explicitly predicted by weather and climate models. This is referred to as the liquid water path (LWP) ($kgm^{-2}$). Other important properties are the cloud base height (CBH) and the cloud top height (CTH), as they impact weather and climate while also capturing the small-scale physics of a cloud.

Our model takes advantage of the fact that cloud fields are spatially structured into cloud layers at specific altitudes (see Figure 1). These cloud layers exhibit different properties and are categorized into various cloud types. In this work, we focus on estimating the physical properties of shallow cumulus clouds, which generally form at altitudes below 2000 m and are the first cloud layer from the ground. These clouds are very difficult to measure using radar and satellite images because their

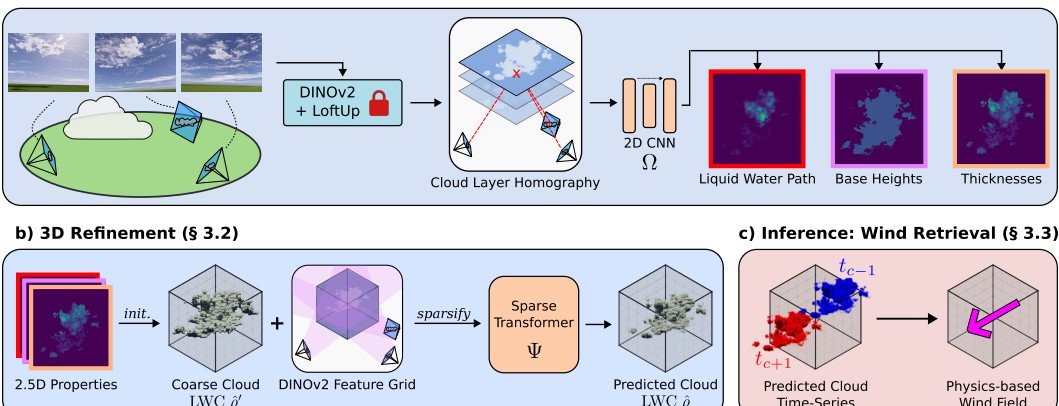

Figure 2: **Model overview.** Our model estimates the liquid water content of clouds using a sparse set of ground-based cameras. **a) Cloud Layer Model:** We leverage an inductive bias on the spatial structure of clouds by defining a homography that maps images to cloud *layers*. The cloud homography is used to predict key 2.5D cloud properties, giving an initial estimate of the 3D cloud layer. **b) 3D Refinement:** A sparse transformer then refines the initial 3D field and estimates the final cloud liquid water content. **c) Inference: Wind Retrieval:** By tracking the motion of cloud reconstructions over time, our method retrieves a height and time-varying horizontal wind profile.

life cycle is short, and each individual cloud is small. In Section 3.1, we show how we can exploit the layer structure of clouds through a homography-guided reconstruction model.

# 3 Method

In this section, we present our approach to estimating a high-resolution 3D grid of cloud properties from ground-based images. We formulate the problem as follows, given $N$ ground-based images $\{I_i\}_{i=1}^N$ together with their corresponding camera poses $\{\boldsymbol{P}_i = [\boldsymbol{R}_i | \boldsymbol{t}_i]\}_{i=1}^N$ and intrinsics $\{\boldsymbol{K}_i\}_{i=1}^N$, the goal is to produce an estimate of the cloud liquid water content at each grid cell $\hat{\rho} \in \mathbb{R}^{N_x \times N_y \times N_z}$.

Figure 2 shows an overview of our method, Cloud4D. We first leverage the spatial structure of clouds and use a homography to map images to cloud layers (Section 3.1). Taking advantage of the vertically thin nature of cloud layers, Cloud4D uses the homography and formulates the cloud layer estimation as an easier 2D-to-2D task. A sparse 3D transformer then refines the predictions using a full learned 3D prior attending over the whole volume (Section 3.2). Lastly, at inference time, we estimate the liquid water content at multiple time steps and extract horizontal wind profiles by tracking our predictions over time (Section 3.3).

## 3.1 Cloud Layer Model

To estimate the properties of a cloud layer, we first consider a homography between an image $I_i$ with homogeneous coordinates $(u_i, v_i, 1)^\top$ and a plane at a height $h$ with world coordinates $(x, y, h)^\top$ as follows:

$$
\begin{bmatrix} x \\ y \\ 1 \end{bmatrix} \sim (\boldsymbol{R}_i - \frac{\boldsymbol{t}_i \boldsymbol{n}^\top}{d}) \boldsymbol{K}_i^{-1} \begin{bmatrix} u_i \\ v_i \\ 1 \end{bmatrix} \tag{1}
$$

where $\boldsymbol{n} = [0, 0, 1]^\top$ and $d = -h$ define the plane as $\boldsymbol{n}^\top \boldsymbol{X} + d = 0$. To search for a cloud layer at an arbitrary height, we consider a feature volume with multiple such planes at H different heights $\{h_i\}_{i=1}^H$. This is similar to previous approaches that build cost volumes for multi-view stereo (Yao et al., 2018; Gu et al., 2020) with the key difference being that our homography is explicitly spatially aligned with cloud layers.

In practice, we leverage the robustness and 3D capabilities (El Banani et al., 2024) of vision foundation models by first processing the images $\{I_i\}_{i=1}^{N}$ using a DINOv2 (Oquab et al., 2023) backbone producing features $\{F_i\}_{i=1}^{N}$. For additional image-level detail, we use LoftUp (Huang et al., 2025) to upsample the DINOv2 features back to the original image resolution and then use a learned projection to downsample the feature dimension to $d_f$ channels.

The DINOv2 features from each camera are lifted to world space using the cloud homography as defined in (1), resulting in a feature volume $V \in \mathbb{R}^{H \times d_f \times N_x \times N_y}$, where the features have been averaged across the N views. Following previous work (Peebles and Xie, 2022; Perez et al., 2018), we use an adaptive layer normalization (adaLN) to condition each of the H total feature layers on the height from which it was sampled.

Due to the vertically thin nature of cloud layers relative to their horizontal length, the main variation in cloud structure will be in the horizontal plane. Motivated by this, we formulate the cloud layer estimation as a 2D-to-2D task, where the 3D cloud layer can be represented using 2.5D cloud properties. Specifically, the feature volume is first flattened into a 2D feature with $H d_f$ channels, such that it can be processed by a 2D CNN, $\Omega$. The 2D CNN predicts key 2D cloud properties, namely the liquid water path LWP, the cloud base heights CBH, and the cloud geometrical thicknesses $\Delta h$.

## 3.2 3D Refinement

In order to obtain a full 3D grid estimate of cloud properties, we lift our 2.5D maps into 3D to give an initial 3D estimate of the liquid water content at each point in space $\hat{\rho}' \in \mathbb{R}^{N_x \times N_y \times N_z}$. To refine this coarse initial estimate, we then use a sparse transformer $\Psi$ which estimates the LWC $\hat{\rho}$ by learning a better 3D distribution of the LWC.

Specifically, we first initialize the 3D LWC $\hat{\rho}'$ from our 2.5D features as the following:

$$\hat{\rho}'^{(x,y,z)} \rightarrow \begin{cases} \frac{\text{LWP}^{(x,y)}}{\Delta h^{(x,y)}} \frac{2(zs_z - \text{CBH}^{(x,y)})}{\Delta h^{(x,y)}}, & \text{if CBH}^{(x,y)} < zs_z < \text{CBH}^{(x,y)} + \Delta h^{(x,y)} \\ 0, & \text{otherwise} \end{cases} \tag{2}$$

where $s_z$ is the size of a voxel along the height dimension. The first term evenly distributes the LWP along the cloud height column, while the second term adds a linear increase towards cloud tops to better match the expected LWC distribution of a cloud (Brenguier et al., 2000).

We then discard empty voxels in our initial LWC $\hat{\rho}'$, extracting a sparse structure of M voxels, where $M \ll N_x N_y N_z$ as the cloud layer is vertically thin relative to the horizontal length. This allows us to process the initial estimate using a sparse transformer, learning a 3D cloud prior without the added computational complexity of attending over a full voxel grid.

To add image-level features to the transformer processing, we concatenate each sparse voxel of LWC with DINOv2 features $\{F_i'\}_{i=1}^{N}$ that are backprojected and averaged across all cameras. As in Section 3.1, the DINOv2 features are spatially upsampled with LoftUp and then downsampled channel-wise using another learned projection to a dimensionality of $d_{f'}$. We additionally add sinusoidal positional encoding based on the coordinates of each sparse voxel.

We normalize the output of the sparse transformer along the height dimension using softmax and then scale each height column such that the original LWP (Section 3.1) is preserved. This formulation takes advantage of the strong 2.5D features from the cloud layer model while learning a full 3D cloud prior.

## 3.3 Wind Retrieval

The large-scale motion of clouds is dominated by horizontal winds that primarily change as a function of altitude. By tracking the movement of our reconstructed clouds, we retrieve the horizontal wind profile over time. This is similar to previous approaches for the estimation of wind through cloud motion using satellites (Horváth and Davies, 2001) with a key difference being that our tracking is done on full 3D LWC fields rather than satellite imagery. This allows us to directly track the motion of clouds along their full vertical width, giving a height-varying horizontal wind profile.

To track the motion of our reconstructed clouds, we visualize the LWC of the reconstructed clouds across different 2D height slices $\rho_{sliced} \in \mathbb{R}^{N_x \times N_y}$ and use an off-the-shelf learned point tracker,

specifically CoTracker3 (Karaev et al., 2024). This approach allows us to use temporal cues to track the reconstructed clouds while also processing long sequences efficiently. By tracking how points on the height-sliced LWC move horizontally over time, the rate of change in pixel space directly gives us horizontal wind retrievals. We provide additional implementation details in the supplemental material.

### 3.4 Implementation Details

**Training** We train our model in two separate stages. The cloud layer model is first trained to predict 2.5D cloud properties, which is then followed by training the sparse transformer for full 3D refinement. For the 2.5D predictions, we optimize for the following objective:

$$\mathcal{L}_{2D} = \mathcal{L}_{\text{LWP}} + \lambda_{\text{CBH}}\mathcal{L}_{\text{CBH}} + \lambda_{\Delta h}\mathcal{L}_{\Delta h} \qquad (3)$$

where all losses are L1 losses applied to each predicted 2.5D cloud property. $\lambda_{\text{CBH}}$ and $\lambda_{\Delta h}$ are hyperparameters to scale the losses to similar ranges and are both set to 0.1.

In the second stage, we freeze the weights trained in the first stage, and train the sparse transformer $\Psi$ with the following objective:

$$\mathcal{L}_{3D} = \|\rho - \hat{\rho}\|_1. \qquad (4)$$

Training is performed for 60k steps in the first stage and 30k steps in the second stage. Optimization is done using Adam (Kingma and Ba, 2015), and takes three days with 4x H100 80GB GPUs.

**Model Configuration** Our model reconstructs volumes from heights of 0 to 4000 m, within a 5 km x 5 km area. We choose a voxel size of 25 m, giving us dimensions of $\hat{\rho} \in \mathbb{R}^{200 \times 200 \times 160}$. For the cloud homography, we sample heights every 200 m starting from 400 m and ending at 3800 m. This results in $H = 18$ heights. The DINOv2 features are downsampled to a channel size of $d_f = d_{f'} = 16$.

## 4 Cloud Datasets

### 4.1 Synthetic Dataset

To train our model, we require ground-based images that are paired with 3D grids representing the cloud liquid water content. There are no instruments that can give such data at the resolution and scale we are aiming for, and thus we rely on realistic large eddy simulations (LES) to generate the necessary training data.

We use the CUDA-accelerated LES software, MicroHH (van Heerwaarden et al., 2017) and simulate three scenarios of cumulus cloud days (ARM (Brown et al., 2002), Cabauw (Tijhuis et al., 2023), and RICO (vanZanten et al., 2011)). By converting our liquid water content from the LES output to scattering coefficients, we render images with Monte Carlo path tracing in Blender's Cycles engine.

For additional data diversity, we also render cloud volumes from Terragen, an application aimed at creating photorealistic natural scenes. As these are not physically realistic and will not accurately represent a real-world cloud liquid water content, we only use this data for pre-training.

For both MicroHH and Terragen volumes, we render images from six different views, with intrinsics and extrinsics matching our real-world dataset (see Section 4.2). For each camera, we render 1500 images from MicroHH volumes and 1000 images from Terragen volumes, totaling 15000 images.

### 4.2 Real-World Cloud Dataset

To evaluate our method, we collect real-world images from six cameras across a two-month period. The six cameras are positioned in an inward-looking array, covering a 5 km x 5 km area where Cloud4D does cloud property estimation. Taking an image every five seconds, the ground-based cameras enable Cloud4D to operate at a high temporal resolution. Comparatively, other instrument-based retrievals, such as satellites, can take up to hours or days. Ground-based cameras, therefore,

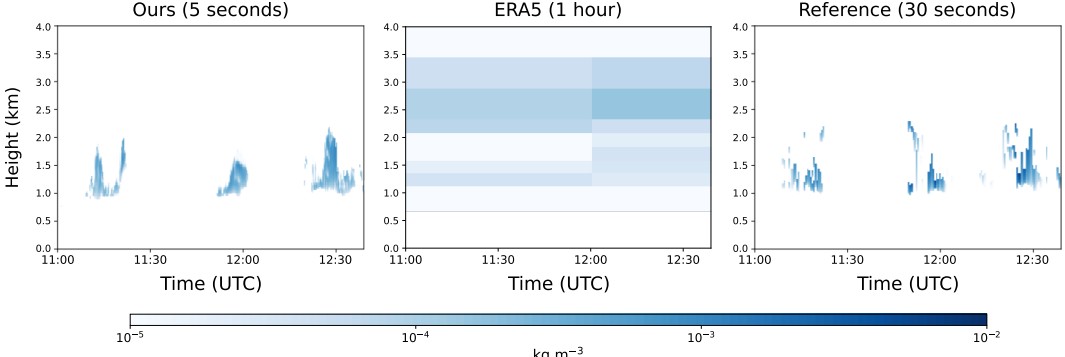

(a) 1D liquid water content along a vertical line aligned with a radar scan

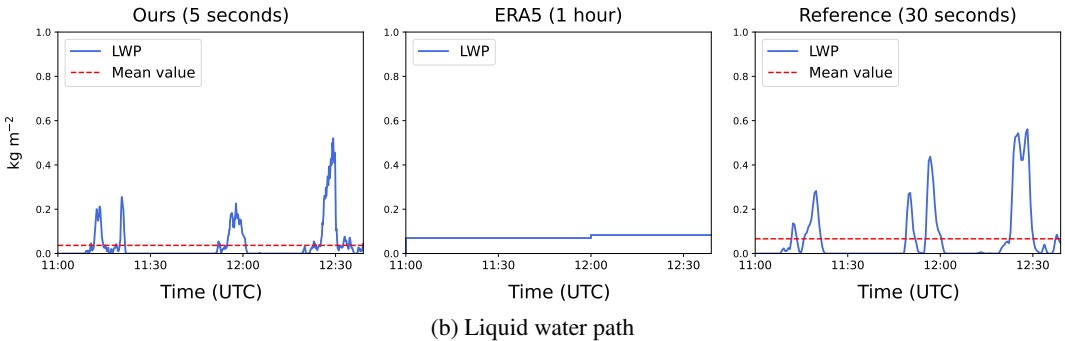

(b) Liquid water path

Figure 3: **Comparison with radar retrievals.** To compare with radar values, we visualize our 3D liquid water content along a single ray over time. ERA5 captures coarse properties such as cloud heights and mean LWP. Comparatively, our method predicts high-resolution cloud properties that match up with radar retrievals.

provide the potential for a cheap and scalable means of obtaining high-resolution observations of clouds at a state-of-the-art spatial and temporal resolution.

To evaluate the accuracy of our cloud property retrievals, the cameras are collocated with instruments that can retrieve liquid water content and horizontal wind profiles. Specifically, a radar retrieves a 1D vertical liquid water content profile above the radar with a 30 m height resolution every 30 seconds. Wind profiles are measured by a radar wind profiler that retrieves the horizontal wind profile every five minutes with a height resolution of 70 m.

Across the two-month deployment, we manually identify 12 days with shallow cumulus clouds, resulting in 17 hours of camera data. This forms the basis of the real-world data that we use for evaluation.

## 5 Experiments

We conduct experiments using our real-world ground-based camera dataset, consisting of 17 hours of camera data capturing shallow cumulus clouds across 12 days. Given that there is no prior work which uses ground-based cameras for the estimation of physical cloud properties, we instead rely on comparisons to methods for satellite-based cloud retrievals (Ronen et al., 2022), radar, wind profiler, and satellite measurements [1], and ERA5 (Hersbach et al., 2020) data [2].

**Radar:** The standard instrument for high-resolution measurements of clouds is generally a cloud-profile radar. Therefore, we compare our cloud property predictions, which span a much larger area

---

[1]Provided by the European Space Agency under the Open Access compliant Creative Commons CC BY-SA 3.0 IGO Licence

[2]Provided by the Copernicus Climate Change Service under the ECMWF Copernicus License.

Table 1: **Quantitative comparison against existing methods**. Cloud occupancy (Occ) is evaluated using an F1 score, where a height column is considered occupied if there are non-zero densities in the column. Mean absolute error is shown for retrievals on liquid water content (LWC), liquid water path (LWP), cloud base height (CBH), and cloud top height (CTH).

| | Occ | LWC ($gm^{-3}$) | LWP ($kgm^{-2}$) | CBH (m) | CTH (m) |
|---|---|---|---|---|---|
| VIP-CT Ronen et al. (2022) | 0.40 | 0.13 | 0.39 | 791.23 | 1021.49 |
| Cloud4D (ours) | **0.70** | **0.03** | **0.06** | **189.58** | **295.77** |

of 5 km x 5 km, with the single height-column retrievals of a radar. To enable this comparison, we evaluate our 3D liquid water content along a single height column at the location of the radar.

**Satellite**: In contrast to cloud-profiling radars, which scan in a limited spatial area, satellites provide retrievals across a large area but are instead limited in temporal resolution. Our method, on the other hand, operates with a significantly larger area compared to cloud-profiling radars, but simultaneously preserves the high temporal resolution, estimating cloud properties every five seconds. The combination of high temporal and spatial resolution is unique to ground-based cameras and is key to resolving the small-scale physics of a cumulus cloud across its life cycle.

**Wind profiler:** Wind can be measured through different instruments, such as satellites, weather balloons, and radar wind profilers. Of these, wind profilers are typically the most accurate, and therefore, we evaluate our wind estimates by comparing them to a wind profiler situated at the site where we collected the evaluation dataset.

### 5.1 Results

Figure 3 compares our cloud liquid water content and cloud liquid water path predictions with radar values. Our predictions have high spatial resolution (25 m) and temporal resolution (5 seconds) while also covering a significantly larger area than the radar. This enables the estimation of fine details across whole clouds and is key to closing the observational gap where current observations do not capture full microphysics across clouds. From Figure 3, we highlight in particular that our predictions of cloud properties, such as cloud base heights, cloud top heights, and liquid water path, closely match the radar retrievals. Global models such as ERA5, instead, operate at kilometer and hourly scales, which do not observe individual clouds but instead capture coarse properties such as the average liquid water path and approximate heights of cloud layers.

Comparing against radar retrievals over a longer period of 12 cumulus days, Table 1 shows quantitative results with our method and also with VIP-CT (Ronen et al., 2022). VIP-CT is a satellite-based cloud retrieval method, which we observe is unable to learn cloud property estimation from ground-based cameras. This is not unexpected, as VIP-CT is designed for satellite imagery and relies on unobstructed orthographic views. Additionally, VIP-CT implicitly learns multi-view geometry, which is significantly more difficult for ground-based cameras where views vary more than for satellites. In comparison, Cloud4D retrieves cloud properties with less than 10% relative error [3] compared to radar values, while doing so in a significantly larger area of 5 km $\times$ 5 km.

Figure 4 shows visualizations of our 3D liquid water content retrievals at times coinciding with satellite imagery at the location of the cameras. We highlight that our cloud envelopes qualitatively match up with satellite retrievals, while operating at a significantly higher temporal resolution of seconds instead of days. This enables cloud property estimation across the full life cycle of a cloud rather than a single snapshot as provided by satellites. We note that our liquid water content retrievals are visualized using Blender with accurate sun positions, enabling coarse qualitative evaluation of cloud heights through the shadow of the clouds.

---

[3]This is calculated using the mean LWC of clouds from the radar GT as following:
$\frac{\text{MAE(LWC of clouds)}}{\text{mean(LWC of clouds from radar GT)}} = \frac{0.029gm^{-3}}{0.321gm^{-3}} = 8.9\%$

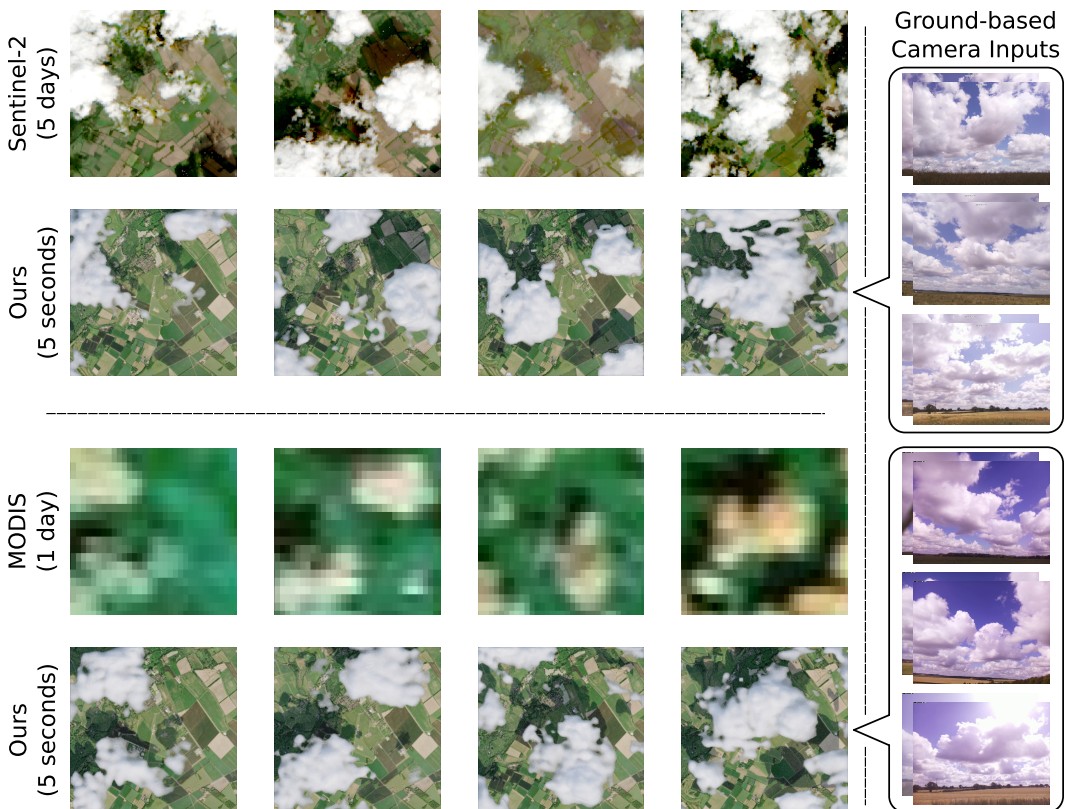

Figure 4: **Qualitative comparison against satellite imagery**. We render our 3D liquid water content predictions from a top-down orthographic view using Blender and compare against 2D image retrievals from Sentinel-2 and MODIS. Cloud4D estimates volumetric cloud properties every five seconds, while on average, Sentinel-2 takes one image every five days and MODIS once per day.

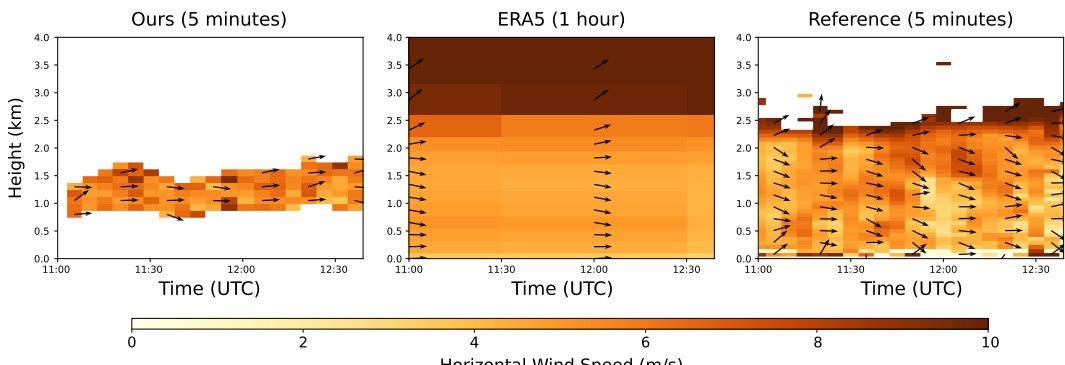

Figure 5: **Comparison of horizontal wind retrieval**. Cloud4D is able to estimate height-varying horizontal wind vectors from the motion of our cloud predictions. Our wind profiles are of similar magnitude and direction to retrievals from a wind profiler. The arrow direction denotes the horizontal wind direction following the convention of a compass bearing.

Figure 5 compares horizontal wind profiles at different heights and times. We observe that our horizontal wind profiles have similar magnitudes and directions when compared to the radar wind profiler retrieval.

## 6 Related Work

**Cloud property estimation**   The retrieval of physical cloud properties is an important task for weather and climate applications. With the advent of deep learning, learned methods for the estimation of cloud properties have been explored using satellite images. Notably, 3DeepCT (Sde-Chen et al., 2021) and VIP-CT (Ronen et al., 2022) have shown comparable retrieval results compared to traditional explicit physics-based methods (Levis et al., 2015, 2017, 2020). However, VIP-CT and 3DeepCT both implicitly learn camera geometries from small-scale synthetic datasets. This is a much more difficult task for ground-based cameras, which will greatly differ in viewing directions compared to the orthographic satellite views. Our method explicitly models the camera geometry by mapping images to cloud layers in world space using a homography, simplifying the cloud property estimation task. Previous works with ground-based cameras have focused on estimating geometric cloud properties such as cloud base heights using stereo cameras (Romps and Öktem, 2018; Öktem et al., 2014), but do not recover physical quantities such as the liquid water content. Our method uses a learned approach to estimate both physical and geometrical properties of clouds, doing so at a high spatial and temporal resolution.

**Weather and climate models**   Recently, there has been a growing number of works on ML-based weather and climate models (ConvLSTM (SHI et al., 2015), GraphCast (Lam et al., 2023), NeuralGCM (Kochkov et al., 2024), Pangu-Weather (Bi et al., 2023), and Aardvark (Allen et al., 2025)). However, these are global models, operating at a kilometer scale, and therefore do not capture the physics of individual clouds. There is a need for higher resolution models such that phenomena that occur at a subgrid scale can be predicted. Our method fills an observational gap, providing high-resolution data enabling the evaluation of fine-scale models and the training of data-driven surrogates.

## 7 Discussion and Limitations

We have trained our model mainly on data of cumulus clouds. We made this choice as cumulus clouds have a large effect on the atmosphere (Chen et al., 2000), while also being difficult to measure with other instruments due to high requirements on temporal resolution and spatial coverage. However, it is worth noting that other cloud types are also significant drivers of the atmosphere (Lee et al., 2021; Chen et al., 2000), which makes the extension of Cloud4D to other cloud types an interesting avenue for future work.

Secondly, we have focused on retrieving cloud properties of a single layer. This is well motivated for ground-based cameras, as any higher layers will generally be occluded. Investigating the retrieval of cloud properties from higher cloud layers would increase the robustness and potential applications of our work and is a promising direction.

Ground-based cameras provide a cheap and scalable means of obtaining cloud retrievals through Cloud4D. However, we note that they also include some limitations that do not apply to other instruments. In particular, they are susceptible to occlusions from environmental conditions such as rain, fog, and snow.

## 8 Conclusion

We introduced Cloud4D, the first learning-based framework that can provide cloud measurements at a high spatial and temporal resolution using ground-based cameras. Leveraging a homography-guided 2D-to-3D transformer and a sparse-voxel refinement stage, Cloud4D reconstructs liquid water content and height-resolved horizontal winds on a 25 m × 25 m × 25 m grid every 5 s. During a two-month deployment, the system achieved an order-of-magnitude finer temporal resolution than state-of-the-art satellite estimates while maintaining < 10% relative error against collocated radar and wind profiler measurements. Because it relies only on low-cost, widely available cameras, Cloud4D offers a path to scalable, high-frequency observations of clouds worldwide. These estimates can close a long-standing observational gap that hampers both physics-based and neural weather and climate models. For future work, extending the framework to multilayer and optically thick cloud regimes, embedding radiative-transfer constraints, and coupling the retrievals with differentiable simulators could further

increase physical fidelity. We plan to release our code, synthetic training data, and 17-hour real-world benchmark dataset.

## Acknowledgements

This work was supported by the Natural Environment Research Council (grant nos.NE/X018539/1 WOEST, NE/X012255/1), ARIA (SCOP-SE01-P06 Next-CAM), an ESPRC scholarship (2922572), and the Royal Society (grant no. URF/R1/191602).

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

## Appendix Table of Contents

# A    Training Details

Table 2: **Overview of our multi-stage training pipeline.** We first pre-train on our synthetic data generated using Terragen, which is then followed by training on our LES data (MicroHH).

| Dataset | Trainable parameters | Learning rate | Schedule | Steps |
|---|---|---|---|---|
| Terragen | 2D CNN | $1 \times 10^{-4}$ | Cosine | 20000 |
| MicroHH | 2D CNN | $1 \times 10^{-5}$ | Constant | 40000 |
| MicroHH | Sparse Transformer | $1 \times 10^{-5}$ | Constant | 30000 |

**Training configurations**    The multi-stage training pipeline is outlined in Table 2. We use the following hyperparameters for all stages of training:

- **Optimizer:** Adam with $(\beta_1, \beta_2) = (0.9, 0.999)$
- **Batch Size:** 1
- **Gradient Clipping:** 1
- **Weight Decay:** 0

**Augmentations**

- **Saturation:** 0.75 to 1.25
- **Hue:** $\frac{-0.05}{3.14}$ to $\frac{0.05}{3.14}$
- **Brightness:** The synthetic cloud images are high-dynamic range (HDR), which we leverage for an augmentation to the brightness. Specifically, we map an $\alpha$-th percentile brightness value to a $\beta$-th percentile brightness value after tonemapping, where the values are randomly sampled as $\alpha \sim \text{Uniform}(0.8, 0.95)$ and $\beta \sim \text{Uniform}(0.7, 0.9)$.
- **Random Dropping of Cameras:** During training of the cloud layer 2D CNN model, we uniformly drop 0 to N-1 input views, where N is the number of camera views.

# B    Model Details

## B.1    Architecture Details

Our 2D CNN implementation is based on the architecture of EDM Karras et al. (2022), with only the input and output channel sizes being changed to fit our task. For the sparse transformer, we follow the architecture of TRELLIS (Xiang et al., 2024), where we do attention after two sparse 3D CNN downsampling blocks, and use channel sizes of 128, 256, and 384. We use 12 sparse transformer blocks, each of which has 12 heads.

## B.2    Cloud Homography Visualization

Figure 6 gives additional intuition into our cloud homography by visualizing the homography at different altitudes. For the visualization, we use RGB images instead of DINOv2 features, where the shown RGB value is averaged across the different views. We note that when the height of the considered plane in the homography matches the cloud layer height, there are minimal artifacts in the feature plane. In comparison, when the height is incorrect, the plane features do not match up across the views, resulting in artifacts when averaged. By considering multiple heights using the cloud homography, our 2D CNN processes the different planes and is able to estimate the correct cloud height.

## B.3    Wind Retrieval Additional Implementation Details

As described in the main paper, Cloud4D retrieves horizontal wind vectors by using CoTracker3 (Karaev et al., 2023) to track our 3D cloud volumes. Here, we give additional details on how this is implemented.

**Pre-processing**    2D slices of our liquid water content predictions are tracked at varying altitudes to retrieve a height-varying wind profile. We find that individual slices can be noisy, and we instead consider the sum of $H = 5$ consecutive slices:

$$\rho_{\text{slice}}^{(t,x,y)} = \sum_{z=h-2}^{h+2} \rho^{(t,x,y,z)}$$

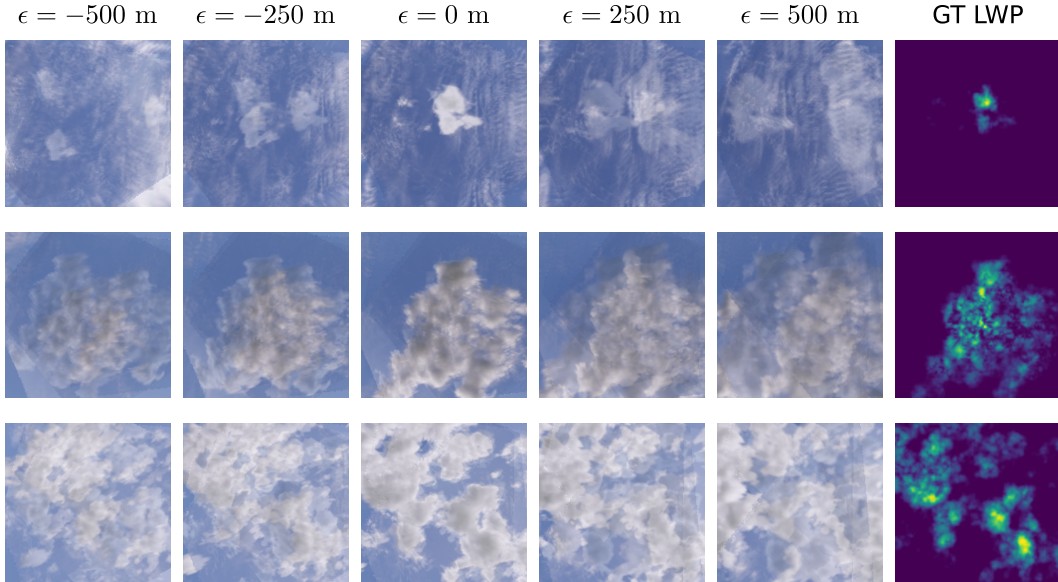

| $\epsilon = -500$ m | $\epsilon = -250$ m | $\epsilon = 0$ m | $\epsilon = 250$ m | $\epsilon = 500$ m | GT LWP |

Figure 6: **Cloud homography visualization**. We visualize the cloud homography by mapping synthetic RGB input images to different heights. $\epsilon$ denotes the height difference from the sampled height and the mean height of the cloud.

where h is the index of the height which we are considering. For each height of the summed liquid water content, we track points across T frames, where T is chosen such that the sequence is long enough for noticeable cloud movement but short enough so that the tracked points are stable. In practice, we choose $T = 20$ frames with 15 seconds between each frame, resulting in tracked points over five minutes.

**Tracking**    For each sequence of liquid water content, we scale the values to a standard grayscale image range of $[0, 1]$, and then initialize points on the first frame for CoTracker3 to track. The points are initialized by uniformly sampling 25 random pixels from the 50th percentile of the highest intensity pixels. This process is repeated for all frames, with all predicted tracks being aggregated into five-minute buckets. This improves the robustness of the wind retrieval but decreases the temporal resolution to five minutes. We extract the wind speed from each track and then take the median wind within each bucket as the final retrieved wind profile. Specifically, for each predicted track that starts at $(x_1, y_1)$ and ends at $(x_2, y_2)$, we estimate the horizontal wind as:

$$u = \frac{s(x_2 - x_1)}{dt}, \quad v = \frac{s(y_2 - y_1)}{dt}$$

where s is the pixel size in meters (25 m) and $dt$ is the duration of the point tracking (five minutes).

**Track Filtering**    We find that there are a significant number of tracks that drift and track empty space. We observe that these failed tracks generally move slowly across the frames. To filter out failed tracked points, we first discard all points that CoTracker3 predicts to be occluded. We then additionally only keep the tracks that have the highest 95% magnitudes in pixel displacement. This removes the failed point tracks as they have lower pixel displacements, and results in a smaller subset of high-quality tracks, which we then use for wind estimation.

Table 3: **LES Configurations.** Voxel sizes and grid extents for all LES cases used for training.

| Case | Voxel sizes (m) | Grid Extent (m) |
|---|---|---|
| Cabauw (Tijhuis et al., 2023) | $10 \times 10 \times 25$ | $5120 \times 5120 \times 4000$ |
| Rico (vanZanten et al., 2011) | $20 \times 20 \times 20$ | $12800 \times 12800 \times 4000$ |
| ARM (Brown et al., 2002) | $10 \times 10 \times 10$ | $6400 \times 6400 \times 4400$ |

## C   Dataset Details

### C.1   LES Settings

The configurations for all cases used for LES training data are given in Table 3. Rendering is done with the native voxel sizes and grid extents shown in the table. To emulate far-away clouds that are outside the simulated grid, the volumes are continuously repeated along the horizontal plane during rendering. For supervision, all voxel sizes are resized to 25 x 25 x 25 using trilinear interpolation, and all grid extents are cropped to the center 5 km x 5 km x 4 km.

### C.2   Rendering

The output of the simulation is a grid of liquid water mixing ratio $q_l$, which needs to be converted to scattering coefficients for rendering. First, the mixing ratio is converted to a droplet concentration $N_d$ by assuming a typical droplet size $r$ for cumulus clouds of 20 $\mu$m (Durbin, 1959):

$$N_d = \frac{3\,q_l}{4\pi\rho_w\,r^3}.$$

where $\rho_w$ is the density of water. The scattering cross-section $\sigma_{\text{scat}}$ and scattering coefficient $\beta$ are then calculated as:

$$\sigma_{\text{scat}} = Q_{\text{scat}}\,\pi r^2, \quad \beta = N_d\,\sigma_{\text{scat}}.$$

where $Q_{\text{scat}}$ is set to 2 for Mie scattering. The scattering probability along a path $\Delta z$ is then given by:

$$P_{\text{scat}} = 1 - \exp(-\beta\,\Delta z).$$

The scattering probability is then used for Monte Carlo path tracing in Blender's Cycles engine.

### C.3   Real-World Dataset

Our camera setup is visualized in Figure 7, and consists of three stereo camera pairs positioned in an inwards-looking triangle. The baselines vary from 190 m to 350 m, while the distances between the pairs are between 5000 m to 8000 m. We note that Cloud4D does not use any stereo vision techniques and thus works on any camera array.

The cameras are synchronized by GPS, such that an image is taken every five seconds. We calibrate the cameras using real-time kinematic positioning, giving a position accurate within a centimeter. Night-time images of stars are then used to optimize for the rotation and focal length.

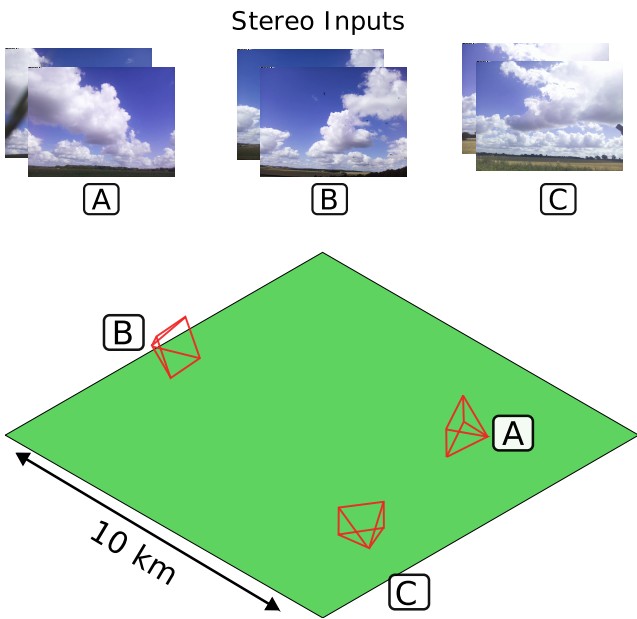

Figure 7: **Camera configuration**. The camera configuration of the real-world dataset is shown with accurate camera poses. For visual clarity, we only illustrate the left camera of each stereo pair. Cloud4D retrieves the cloud liquid water content in the middle 5 km x 5 km area.

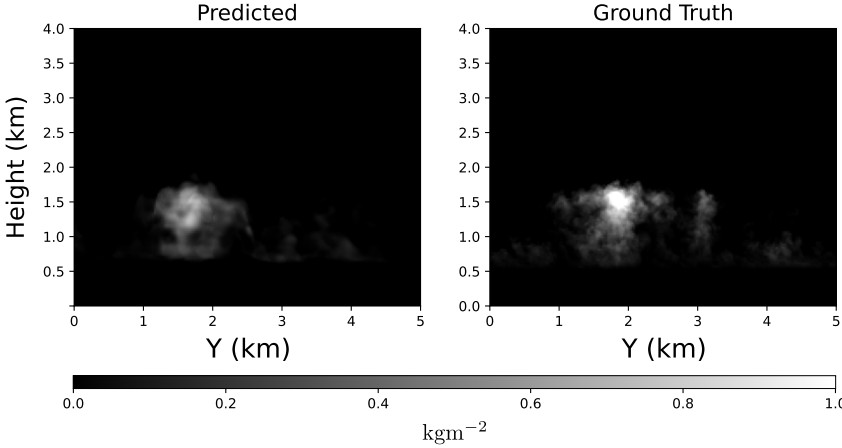

(a) LWC summed along the x-axis

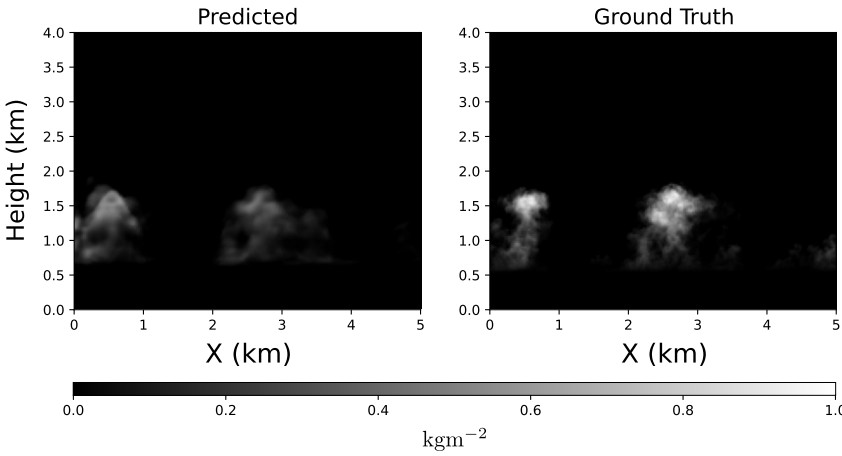

(b) LWC summed along the y-axis

Figure 8: **Synthetic data evaluation.** We visualize our liquid water content predictions by summing along (a) the x-axis and (b) the y-axis. Our predictions capture the cloud envelope, including the cloud base height, cloud top height, and the plume shape. We note that the evaluated data is from an unseen LES configuration.

# D   Additional Experiment: LES Data

The evaluation in the main paper has relied on comparisons to radar and satellite data. However, the radar retrievals only evaluate a narrow vertical beam, and satellite retrievals generally focus on a top-down photometric evaluation. To evaluate the side profiles of our predictions, we generate an additional LES-generated volume using a configuration unseen during training (BOMEX (Siebesma et al., 2003)). Figure 8 shows the liquid water content summed along the x-axis and the y-axis. Our predictions retrieve accurate cloud base heights and cloud top heights, while also capturing correct plume shapes.

# E   Additional Results

## E.1   Ablations

In Table 4, we show an ablation study on the sparse transformer and the number of input views evaluated using radar retrievals on our real-world datasets.

**Sparse transformer**    The addition of the sparse transformer improves predictions on all metrics (liquid water content, cloud base heights, and cloud top heights). We note that the sparse transformer

Table 4: Ablation study on the sparse transformer and the number of input views. We note that the liquid water path and cloud occupancy remain unchanged with the sparse transformer, as it preserves the original liquid water path. For the ablation on input views, we uniformly drop camera pairs from the input.

| Sparse Transformer | | | | | |
|---|---|---|---|---|---|
| **Model** | **Occ** | **LWC** ($gm^{-3}$) | **LWP** ($kgm^{-2}$) | **CBH (m)** | **CTH (m)** |
| 2D CNN Only | 0.70 | 0.030 | 0.063 | 202.04 | 299.04 |
| + Sparse Transformer | **0.70** | **0.029** | **0.063** | **189.58** | **295.77** |

| Input Views | | | | | |
|---|---|---|---|---|---|
| **Dropped Views** | **Occ** | **LWC** ($gm^{-3}$) | **LWP** ($kgm^{-2}$) | **CBH (m)** | **CTH (m)** |
| 0 | 0.70 | **0.029** | **0.063** | **189.58** | **295.77** |
| 2 | **0.76** | 0.038 | 0.070 | 316.28 | 371.71 |
| 4 | 0.68 | 0.037 | 0.072 | 325.03 | 383.16 |

Table 5: Real-world performance variation across different cloud properties. We use the GT radar retrievals to classify the cloud properties in our real-world dataset. Note that the cloud thickness refers to the vertical geometric thickness of the cloud.

| **Cloud Coverage** | **Occ** | **LWC** ($gm^{-3}$) | **LWP** ($kgm^{-2}$) | **CBH (m)** | **CTH (m)** |
|---|---|---|---|---|---|
| 0.20 - 0.45 | 0.65 | 0.034 | 0.077 | **149.36** | **287.12** |
| 0.75 - 0.90 | **0.75** | **0.023** | **0.048** | 230.92 | 304.67 |

| **Mean CBH (m)** | **Occ** | **LWC** ($gm^{-3}$) | **LWP** ($kgm^{-2}$) | **CBH (m)** | **CTH (m)** |
|---|---|---|---|---|---|
| 800 - 1350 | **0.74** | **0.027** | **0.055** | 198.63 | **281.29** |
| 1350 - 1650 | 0.63 | 0.033 | 0.077 | **173.36** | 321.73 |

| **Mean Cloud Thickness (m)** | **Occ** | **LWC** ($gm^{-3}$) | **LWP** ($kgm^{-2}$) | **CBH (m)** | **CTH (m)** |
|---|---|---|---|---|---|
| 150 - 275 | 0.69 | **0.025** | **0.050** | 194.84 | **269.58** |
| 275 - 500 | **0.71** | 0.033 | 0.076 | **184.09** | 323.12 |

preserves the liquid water path from the 2D CNN prediction and therefore will not affect the evaluation of the liquid water path and the cloud occupancy.

**Number of input views** The estimation of CBH and CTH worsens as the number of views decreases, but the LWC and the LWP are less affected. The decreasing CBH and CTH accuracy with dropped views matches established results in 3D vision, where increasing the number of views improves accuracy in 3D shape estimation.

### E.2 Performance Across Different Cloud Properties

Table 5 shows results across different cloud conditions where the radar GT has been used to classify the cloud coverage, mean cloud base height, and mean cloud geometric thickness, over one-hour segments in our real-world dataset. From the results, we note slightly worse performance for higher cloud base heights and also geometrically thicker clouds, but not to a significant extent.

### E.3 Additional Real-World Dataset Radar Examples

In Figures 9 to 13 we show additional comparisons to radar retrievals.

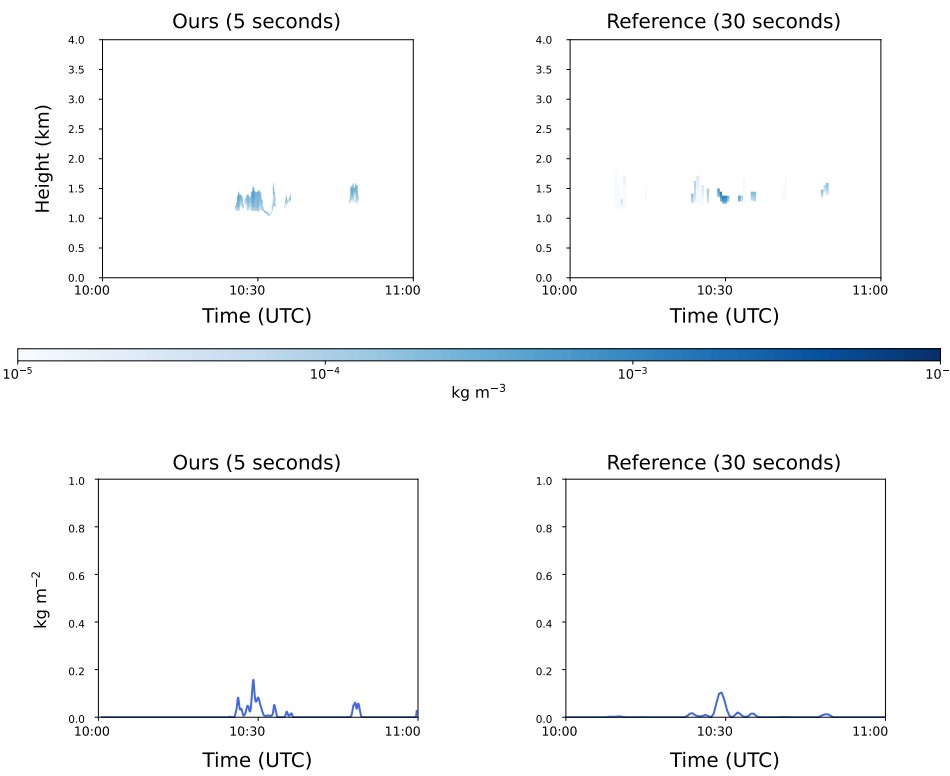

Figure 9: **Example 1.** Additional results comparing the liquid water content and liquid water path with radar retrievals.

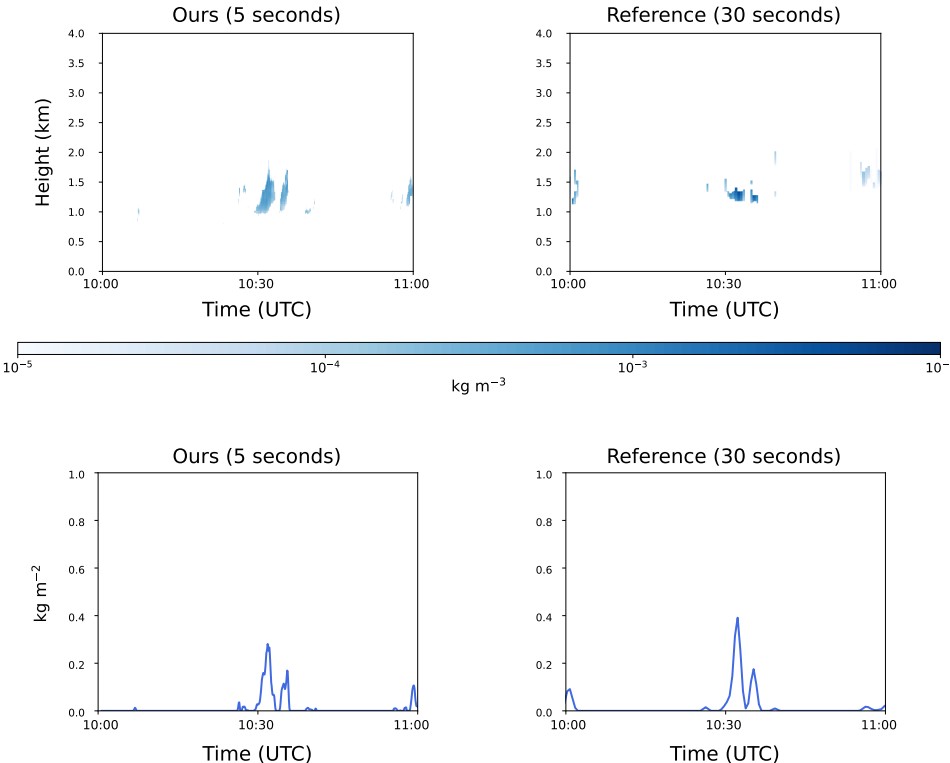

Figure 10: **Example 2.** Additional results comparing the liquid water content and liquid water path with radar retrievals.

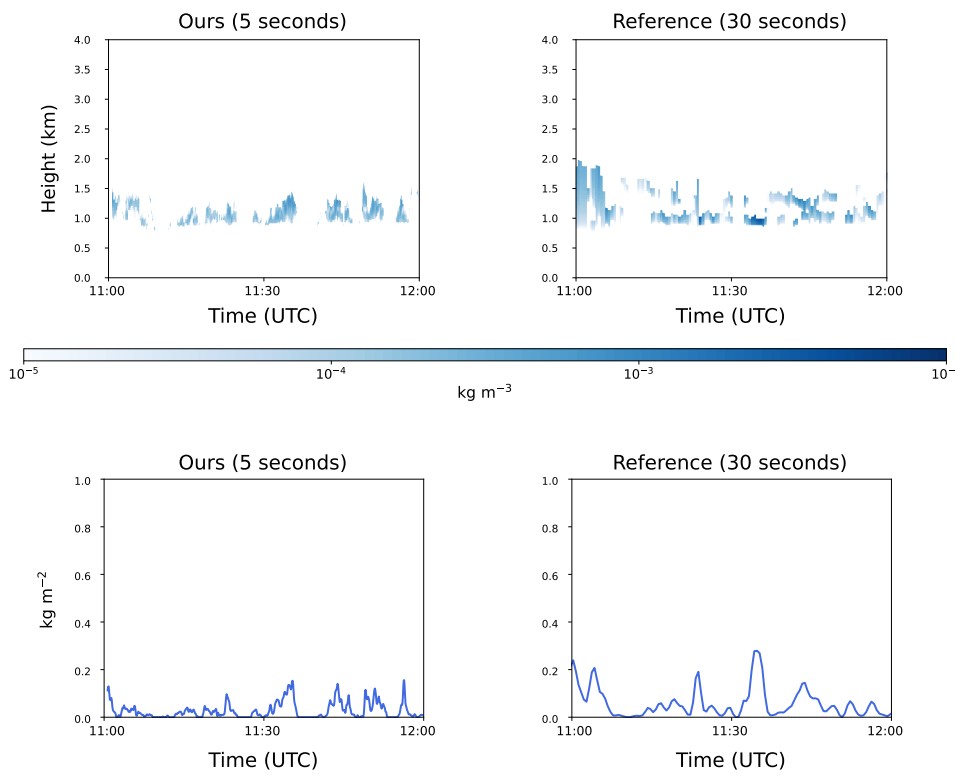

Figure 11: **Example 3.** Additional results comparing the liquid water content and liquid water path with radar retrievals.

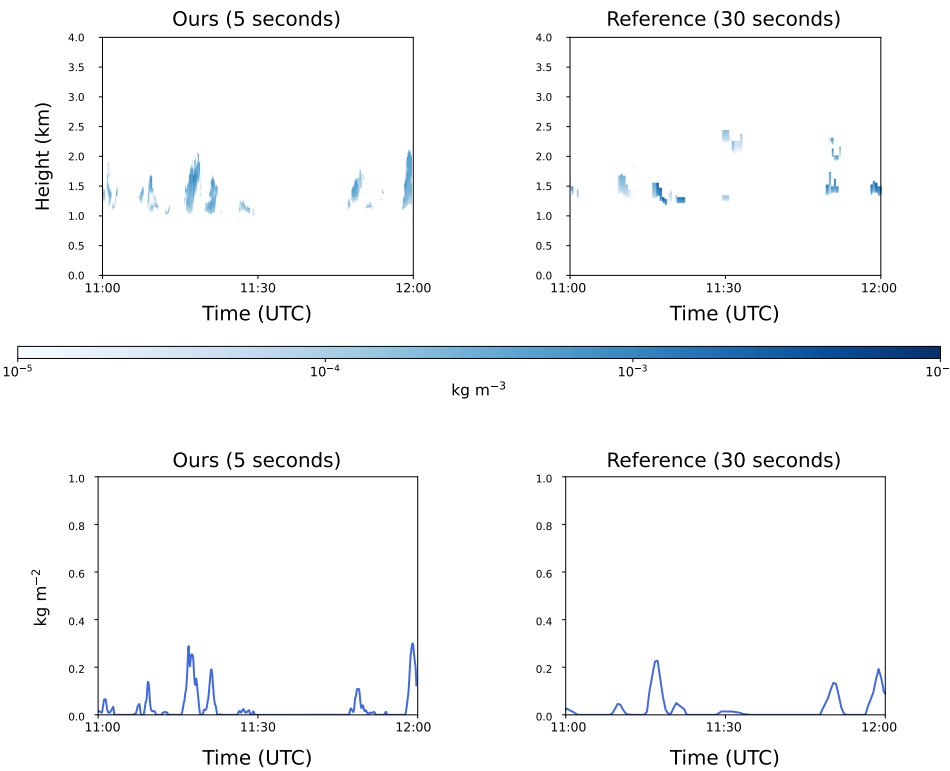

Figure 12: **Example 4.** Additional results comparing the liquid water content and liquid water path with radar retrievals.

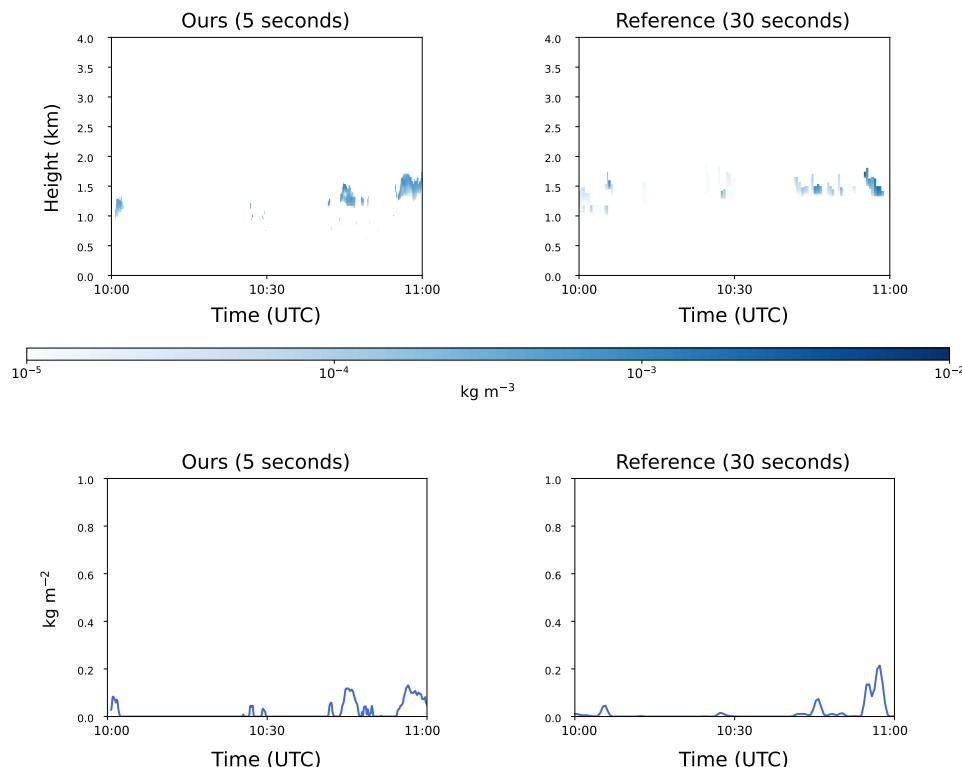

Figure 13: **Example 5.** Additional results comparing the liquid water content and liquid water path with radar retrievals.

## Appendix References

W. G. Durbin. Droplet sampling in cumulus clouds. *Tellus*, 11(2):202–215, 1959.

Tero Karras, Miika Aittala, Timo Aila, and Samuli Laine. Elucidating the design space of diffusion-based generative models. In *Proc. NeurIPS*, 2022.

A. Pier Siebesma, Christopher S. Bretherton, Andrew Brown, Andreas Chlond, Joan Cuxart, Peter G. Duynkerke, Hongli Jiang, Marat Khairoutdinov, David Lewellen, Chin-Hoh Moeng, Enrique Sanchez, Bjorn Stevens, and David E. Stevens. A large eddy simulation intercomparison study of shallow cumulus convection. *Journal of the Atmospheric Sciences*, 60(10):1201 – 1219, 2003.

Jianfeng Xiang, Zelong Lv, Sicheng Xu, Yu Deng, Ruicheng Wang, Bowen Zhang, Dong Chen, Xin Tong, and Jiaolong Yang. Structured 3d latents for scalable and versatile 3d generation. *arXiv preprint arXiv:2412.01506*, 2024.

