# OpenReview forum: "Cloud4D: Estimating Cloud Properties at a High Spatial and Temporal Resolution"
_NeurIPS.cc/2025/Conference — NeurIPS 2025 spotlight_

### Official Review · Reviewer_fVbw · 2025-06-19

**Clarity:** 2
**Significance:** 2
**Originality:** 1
**Rating:** 3
**Confidence:** 4

**Summary:**

The paper introduces Cloud4D, a novel learning-based framework designed to estimate cloud properties at high spatial and temporal resolutions using synchronized ground-based cameras. The framework leverages a homography-guided 2D-to-3D transformer architecture to predict the full spatial distribution of liquid-water content and horizontal wind vectors. Through a two-month real-world deployment with six cameras, Cloud4D demonstrates a significant improvement in space-time resolution compared to state-of-the-art satellite measurements while maintaining single-digit relative error against collocated radar measurements. In summary, the paper presents a groundbreaking approach to cloud property estimation that addresses the limitations of traditional methods and offers a scalable solution for high-resolution cloud observations

**Questions:**

1. The authors mention that Cloud4D is primarily trained on cumulus clouds and focuses on single-layer cloud properties. While this is well-motivated for ground-based cameras, the limitation to specific cloud types and single-layer clouds restricts the framework's applicability. Could the authors elaborate on the challenges and potential strategies for extending Cloud4D to other cloud types (e.g., stratocumulus, cirrus) and multi-layer clouds?  How might the framework handle occlusions and distinguish between different cloud layers?
﻿
2. The authors should consider providing error bars or other statistical measures for the reported metrics. Additionally, explaining the factors contributing to the variability (e.g., train/test splits, initialization) would strengthen the statistical significance of the findings.
﻿
3. The paper compares Cloud4D with radar, satellite, and other instruments, but the comparisons with existing machine learning-based cloud retrieval methods are somewhat limited.How does the explicit modeling of camera geometry in Cloud4D contribute to its performance compared to the implicit learning approaches used?
﻿
4. The paper does not include a detailed theoretical analysis of the proposed framework, such as the assumptions made in the homography-guided 2D-to-3D transformation and the limitations of the sparse-voxel refinement approach.
﻿
5. The authors could include experiments or simulations to validate the framework's performance under various environmental conditions. How does Cloud4D handle partially obscured views or low visibility conditions? Are there any preprocessing steps or modifications to the framework that can mitigate the impact of these conditions?

**Ethical Concerns:**

["NO or VERY MINOR ethics concerns only"]

**Limitations:**

yes

**Paper Formatting Concerns:**

I did not find any major formatting issues.

**Quality:**

2

**Strengths And Weaknesses:**

### Strengths:
﻿
- The paper is technically sound and presents a novel framework (Cloud4D) that effectively addresses the challenge of estimating cloud properties at high spatial and temporal resolutions. The proposed method demonstrates significant improvements over existing satellite measurements, achieving an order-of-magnitude enhancement in temporal resolution while maintaining high accuracy (less than 10% relative error) when compared to radar measurements.
- The authors provide comprehensive experimental validation, including comparisons with radar, satellite, and wind profiler data, which strengthens the credibility of their claims. The use of a two-month real-world deployment with six cameras further supports the robustness of the framework.

### Weaknesses:
﻿
- While the experimental results are promising, the paper does not provide error bars or confidence intervals for the reported metrics.
﻿
- Some sections could benefit from additional details. For example, the implementation details of the wind retrieval method could be elaborated further to provide a clearer understanding of how the horizontal wind vectors are estimated from the reconstructed cloud fields.
﻿
- While the combination of techniques presented in the paper is novel, some components of the framework, such as the use of vision foundation models (e.g., DINOv2) and sparse transformers, have been explored in other computer vision applications.

---

> ### Author Rebuttal · Authors · 2025-07-31
>
> We thank the reviewer for their feedback and appreciate that they find our approach "groundbreaking" and the evaluation to be comprehensive. Please find below our response to your comments.
>
> &nbsp;
>
> >Weakness 1. While the experimental results are promising, the paper does not provide error bars or confidence intervals for the reported metrics
>
> The main source of randomness in our experiments comes from the seed used for the weight initialization and dataset sampling during training. The model takes over 280 H100 hours to train, so running multiple full training runs is not feasible. To test the model variability, we have included results below on the mean and the 2-sigma error over five runs with random seeds trained for 9k steps (the full training run takes 90k steps). The results show that our method is robust to randomness in initialization and data sampling.
>
> | &nbsp; &nbsp; &nbsp; &nbsp; Occ &nbsp; &nbsp; &nbsp; &nbsp; | &nbsp; &nbsp; &nbsp; LWC (gm⁻³) &nbsp; &nbsp; &nbsp; | &nbsp; &nbsp; &nbsp; LWP (kgm⁻³) &nbsp; &nbsp; &nbsp; | &nbsp; &nbsp; &nbsp; &nbsp; CBH (m) &nbsp; &nbsp; &nbsp; &nbsp; | &nbsp; &nbsp; &nbsp; &nbsp; CTH (m) &nbsp; &nbsp; &nbsp; &nbsp; |
> | :---: | :---: | :---: | :---: | :---: |
> | 0.53 $\pm$ 0.06 | 0.0319  $\pm$ 0.018| 0.0764 $\pm$ 0.0008 | 160.81 $\pm$ 15.32 | 285.20 $\pm$ 39.34 |
>
> &nbsp;
>
> >Weakness 2. the implementation details of the wind retrieval method could be elaborated further.
>
> Additional implementation details of the wind retrieval method are given in Appendix B.3. We are happy to expand on this if there is anything missing. We will also release our code, which includes the full implementation details.
>
> &nbsp;
>
> >Weakness 3. While the combination of techniques presented in the paper is novel, some components of the framework, such as the use of vision foundation models (e.g., DINOv2) and sparse transformers, have been explored in other computer vision applications
>
> We thank the reviewer for acknowledging the novelty of our proposed system. We agree that foundational models like DINOv2 and sparse transformers are established tools. Indeed, our motivation was precisely to leverage their representational power to solve the challenging task of estimating 3D cloud properties from 2D images, which has not been done before.
>
> &nbsp;
>
> >Q1. Could the authors elaborate on the challenges and potential strategies for extending Cloud4D to other cloud types (e.g., stratocumulus, cirrus) and multi-layer clouds?
>
> As most cloud fields follow a vertically thin and horizontally wide structure, our method can be applied to other cloud types too. This could be accomplished by generating synthetic data for other cloud types.
>
> While our cloud layer estimation model is currently designed for single cloud layers, it could be extended to multiple cloud layers. For example, instead of predicting 2.5D cloud features $\mathbb{R}^{N_x \times N_y \times 3}$, you could estimate additional channels to represent higher layers $\mathbb{R}^{N_x \times N_y \times 3L}$, where L would be the number of cloud layers to predict and would realistically not need to be more than two or three.
>
> &nbsp;
>
> >Q2. How might the framework handle occlusions and distinguish between different cloud layers?
>
> Our current model is designed for single cloud layers. See our comment to Q1 above for suggestions on how this could be extended.
>
> &nbsp;
>
> >Q3. The authors should consider providing error bars or other statistical measures for the reported metrics
>
> See answer to Weakness 1 above.
>
> &nbsp;
>
> >Q4. Explaining the factors contributing to the variability (e.g., train/test splits, initialization) would strengthen the statistical significance of the findings
>
> **Train/test splits.** To ensure minimal leakage between training and test sets for the synthetic part of the evaluation, we do the split based on different meteorological environments used to configure the LES-based generation of synthetic cloud data.
>
> For training, we use the following three meteorological conditions: ARM [1], Cabauw [2], and RICO [3].
>
> For testing, we use BOMEX [4].
>
> **Initialization.** We initialize our model weights using a Xavier uniform distribution.
>
> &nbsp;
>
> >Q5. How does the explicit modeling of camera geometry in Cloud4D contribute to its performance compared to the implicit learning approaches used?
>
> Our camera setup consists of sparse views that are spaced far apart. As a result, the input images from the different views will greatly differ. The explicit modelling of camera geometry makes it easier to learn how to predict full 3D cloud properties from the sparse input images.
>
> &nbsp;
>
> >Q6. Assumptions made in the homography-guided 2D-to-3D transformation and the limitations of the sparse-voxel refinement approach
>
> Find below a list of assumptions in our model:
>
> (1) The homography assumes a pinhole camera model with no radial distortion and accurate camera poses. These are common assumptions for 3D vision tasks.
>
> (2) The backprojection of DINOv2 features assumes the cloud layer being reconstructed is not occluded. This is generally not a problem for single cloud layers, as the cameras are upwards facing, such that occlusions are minimized. Additionally, while one camera might be partially occluded, the features from other unoccluded cameras will help average out errors.
>
> (3) The sparse transformer only refines in local areas of the estimated cloud from the cloud layer estimation model.
>
> &nbsp;
>
> >Q7. The authors could include experiments or simulations to validate the framework's performance under various environmental conditions
>
> We have investigated our model's performance across different conditions in our real-world dataset, including different numbers of camera views, cloud coverages, cloud base heights, and cloud geometric thicknesses. These results are shown below, where the radar GT has been used to classify the cloud coverage, mean cloud base height, and mean cloud geometric thickness, over one-hour segments in our real-world dataset.
>
> From these results, we highlight that the model’s performance slightly worsens for higher cloud base heights and also geometrically thicker clouds, but not to a significant extent.
>
> | Cloud Coverages &nbsp; &nbsp; &nbsp; &nbsp; &nbsp; &nbsp; | &nbsp; &nbsp; Occ &nbsp; &nbsp; | &nbsp; LWC (gm⁻³) &nbsp; | &nbsp; LWP (kgm⁻³) &nbsp; | &nbsp; &nbsp; CBH (m) &nbsp; &nbsp; | &nbsp; &nbsp; CTH (m) &nbsp; &nbsp; |
> | :--- | :---: | :---: | :---: | :---: | :---: |
> | 0.20 - 0.45 | 0.65 | 0.034 | 0.077 | **149.36** | **287.12** |
> | 0.75 - 0.90 | **0.75** | **0.023** | **0.048** | 230.92 | 304.67 |
>
> &nbsp;
>
> | Mean CBH (m) &nbsp; &nbsp; &nbsp; &nbsp; &nbsp; &nbsp; &nbsp; &nbsp; | &nbsp; &nbsp; Occ &nbsp; &nbsp; | &nbsp; LWC (gm⁻³) &nbsp; | &nbsp; LWP (kgm⁻³) &nbsp; | &nbsp; &nbsp; CBH (m) &nbsp; &nbsp; | &nbsp; &nbsp; CTH (m) &nbsp; &nbsp; |
> | :--- | :---: | :---: | :---: | :---: | :---: |
> | 800 - 1350 | **0.74** | **0.027** | **0.055** | 198.63 | **281.29** |
> | 1350 - 1650 | 0.63 | 0.033 | 0.077 | **173.36** | 321.73 |
>
> &nbsp;
>
> | Mean Cloud Thickness (m) &nbsp; &nbsp; &nbsp; | &nbsp; &nbsp; Occ &nbsp; &nbsp; | &nbsp; LWC (gm⁻³) &nbsp; | &nbsp; LWP (kgm⁻³) &nbsp; | &nbsp; &nbsp; CBH (m) &nbsp; &nbsp; | &nbsp; &nbsp; CTH (m) &nbsp; &nbsp; |
> | :--- | :---: | :---: | :---: | :---: | :---: |
> | 150 - 275 | 0.69 | **0.025** | **0.050** | 194.84 | **269.58** |
> | 275 - 500 | **0.71** | 0.033 | 0.076 | **184.09** | 323.12 |
>
> &nbsp;
>
> >Q8. How does Cloud4D handle partially obscured views or low visibility conditions?
>
> **Partially obscured views.** To evaluate how Cloud4D might be affected by partially obscured views,  we have included results below on how Cloud4D is affected by uniformly dropping random camera pairs during evaluation. The results show that predictions for cloud base heights and cloud top heights worsen as views are dropped, but that liquid water content and liquid water path predictions are less affected. The decreasing CBH and CTH accuracy with missing views matches established results in 3D vision, where increasing the number of views improves accuracy in 3D shape estimation.
>
> | Number of Dropped Views &nbsp; &nbsp; &nbsp; | &nbsp; &nbsp; Occ &nbsp; &nbsp; | &nbsp; LWC (gm⁻³) &nbsp; | &nbsp; LWP (kgm⁻³) &nbsp; | &nbsp; &nbsp; CBH (m) &nbsp; &nbsp; | &nbsp; &nbsp; CTH (m) &nbsp; &nbsp; |
> | :--- | :---: | :---: | :---: | :---: | :---: |
> | 0 | 0.70 | **0.029** | **0.063** | **189.58** | **295.77** |
> | 2 | **0.76** | 0.038 | 0.070 | 316.28 | 371.71 |
> | 4 | 0.68 | 0.037 | 0.072 | 325.03 | 383.16 |
>
> **Low visibility conditions.** Our current implementation focuses on cases where no rain or fog is obscuring the views. Mitigating the impact of these conditions is left to future work (see Q9 for suggestions).
>
> &nbsp;
>
> >Q9. Are there any preprocessing steps or modifications to the framework that can mitigate the impact of these conditions?
>
> Yes, here are some suggestions:
>
> (1) The model could directly learn to be robust to low-visibility conditions by including such cases in the training data. The simulation of rain or fog could be implemented using existing techniques in Blender.
>
> (2) The impact of low visibility conditions could be mitigated through preprocessing by using weather data to filter out data with rain or fog.
>
> &nbsp;
>
> [1] Large-eddy simulation of the diurnal cycle of shallow cumulus convection over land, Brown et al. (Royal Meteorological Society 2002)
>
> [2] An efficient parameterization for surface shortwave 3d radiative effects in large-eddy simulations of shallow cumulus clouds, Tijhuis et al. (Advances in Modeling Earth Systems 2023)
>
> [3] Controls on precipitation and cloudiness in simulations of trade-wind cumulus as observed during rico, vanZanten et al. (Advances in Modeling Earth Systems 2011)
>
> [4] A large eddy simulation intercomparison study of shallow cumulus convection, Siebesma et al. (Atmospheric Sciences 2003)

---

> > ### Author Response · Authors · 2025-08-07
> >
> > >How does the explicit modeling of camera geometry in Cloud4D contribute to its performance compared to the implicit learning approaches used?
> >
> > To further address this question, in our latest response to reviewer e4kC, we evaluate our model and VIP-CT on a different test site and camera setup. The results are copied here:
> >
> >
> > | &nbsp; &nbsp; &nbsp;  | &nbsp; &nbsp; Occ &nbsp; &nbsp; | &nbsp; LWC (gm⁻³) &nbsp; | &nbsp; LWP (kgm⁻³) &nbsp; | &nbsp; &nbsp; CBH (m) &nbsp; &nbsp; | &nbsp; &nbsp; CTH (m) &nbsp; &nbsp; |
> > | :--- | :---: | :---: | :---: | :---: | :---: |
> > | VIP-CT &nbsp; &nbsp; | 0.15 | 8.503 | 28.907 | 2257.44 | 1969.44 |
> > | Ours &nbsp; &nbsp; | **0.64** | **0.019** | **0.037** | **235.74** | **346.82** |
> >
> > &nbsp;
> >
> > The results show that our method generalises to other camera setups without retraining, whereas VIP-CT does not. This is an inherent limitation of the implicit learning approaches which will overfit to the training camera setup. Our homography guided approach, projects the features to world space making the cloud volume prediction largely independent of the camera poses.
> >
> > Please let us know if there are any remaining questions.

---

### Official Review · Reviewer_oz8q · 2025-06-24

**Clarity:** 3
**Significance:** 4
**Originality:** 4
**Rating:** 6
**Confidence:** 3

**Summary:**

This paper presents a scheme for 4D cloud estimation (3D cloud with time). In the first (course)-level, they leverage homography and in the second-level, they trained a model for more fine-grained cloud prediction. The authors use simulation for training and collect their own real-world datasets for validation. Previous work either uses implicit geometry or focus solely on cloud base heights, and this is the first work that scale to real-world scenarios.

**Questions:**

**Dataset Generation**:
1. Could the authors elaborate on the synthetic dataset generation process?
2. How was the data split between training and test sets?
3. Please clarify the "three scenarios" mentioned in Line 180.
4. Were different lighting conditions (such as sun azimuth) considered, and was lighting augmentation performed?
5. How accurately does the LES simulation reflect real-world conditions, and is this a standard methodology?
6. What types of real-world clouds cannot be simulated using cloud simulation?
7. It would be beneficial to include visualizations comparing synthetic and real-world data.

**Training and Fine-tuning**:
1. Has consideration been given to using Sentinel-2 imagery for model fine-tuning?
2. Given the two-month real-world dataset and Sentinel-2's average 5-day revisiting time, it would be valuable to explore whether real-world fine-tuning improves performance.
3. Does scaling up the training dataset size help?

**Quantitative Evaluation**:
1. The paper lacks comprehensive quantitative evaluation of real-world data performance.

**Alternative Approaches**:
1. Have alternative methods such as NERF or DUST3R been considered for 3D estimation?
2. How do these methods compare to the proposed approach?

**Ethical Concerns:**

["NO or VERY MINOR ethics concerns only"]

**Final Justification:**

The authors answered my questions and resolved my concerns. I recommend an acceptation of this work.

**Limitations:**

Yes.

**Quality:**

4

**Strengths And Weaknesses:**

Strength:
1. This paper is well-written and the idea is brought forward pretty neat and clear.
2. This is a very interesting cloud estimation problem and the authors move beyond 3D geometry (like CBH or CTH) and include LWC, LWP. This allows for precise estimation of the Earth atmosphere at a very fine-grained estimation.
3. The idea of validation from a bird-eye view using Sentinel-2 and MODIS is interesting and helpful to validate this scheme and the results are pretty astonishing to me - Cloud prediction and shadow estimation have been a long under-explored problem and this piece of work can offer as a new labeling technique.

Weakness:
1. I think the comparison is not thorough and it would be helpful to include methods from other domains. But this does not compromise the quality and significance of this work

---

> ### Author Rebuttal · Authors · 2025-07-31
>
> We thank the reviewer for their feedback and are happy to hear that they find the cloud estimation task very interesting and that they were impressed with the results in our satellite-based comparison. Please find below our response to your comments.
>
> &nbsp;
>
> ### Dataset Generation
>
> >Q1. How was the data split between training and test sets?
>
> For training, we use high-resolution LES, which is a state-of-the-art simulation approach for clouds. To ensure that our model generalizes across different conditions, we simulate a number of cases with different meteorological environments. In practice, we use a set of well-studied meteorological conditions and refer to these as “scenarios”.
>
> To ensure minimal leakage between training and test sets for the synthetic part of the evaluation, we do the split based on the scenarios.
>
> For training, we use the three scenarios mentioned in Line 180 of the paper: ARM [1], Cabauw [2], and RICO [3].
>
> For testing, we use: BOMEX [4]
>
> >Q2. Please clarify the "three scenarios" mentioned in Line 180.
>
> See answer to Q1 above.
>
> >Q3. Were different lighting conditions (such as sun azimuth) considered, and was lighting augmentation performed?
>
> Yes, during the data generation, we randomly sample the sun position. We additionally sample the air density and the dust density, both of which also affect the lighting conditions.
>
> Image augmentations are applied during training, where we vary the saturation, hue, and brightness. For full details, we refer to the training details that can be found in Appendix A of the supplemental material.
>
> >Q4. How accurately does the LES simulation reflect real-world conditions, and is this a standard methodology?
>
> Yes, LES is a standard methodology for the simulation of high-resolution clouds. It is commonly used for the study of cloud development, and the simulations typically agree well with real-world measurements for cases with shallow convection (as in our setting) [4].
>
> >Q5. What types of real-world clouds cannot be simulated using cloud simulation?
>
> There are no specific limits to the kinds of clouds that can be simulated. However, as the microphysics (behaviour of droplets and ice crystals) cannot be simulated directly, clouds involving complex precipitation processes or a mixture of liquid and ice are more challenging to simulate. We focus on non-precipitating shallow cumulus, which are commonly studied through simulation.
> >Q6. It would be beneficial to include visualizations comparing synthetic and real-world data.
>
>
> Thank you for the suggestion. We will add an example of this to the supplementary.
>
> &nbsp;
>
> ### Training and Fine-tuning
>
> >Q7. Has consideration been given to using Sentinel-2 imagery for model fine-tuning? Given the two-month real-world dataset and Sentinel-2's average 5-day revisiting time, it would be valuable to explore whether real-world fine-tuning improves performance.
>
> Even with a 5-day revisiting time, Sentinel-2 only visits the area ~12 times over the two months our cameras were deployed. Of these visits, some are completely clear skies, and others are dominated by non-cumulus clouds. Unfortunately, this means that there is not enough data for fine-tuning at the current stage.
>
> >Q8. Does scaling up the training dataset size help?
>
> Yes, from initial experiments, we found that increasing the number of scenarios used for training helped with the robustness of our model. This is likely due to the increased diversity in meteorological conditions in our training data resulting in better generalization.
>
> &nbsp;
>
> ### Quantitative Evaluation
>
> >Q9. The paper lacks comprehensive quantitative evaluation of real-world data performance.
>
> Table 1 in the paper shows quantitative metrics compared to radar retrievals using our two-month real-world dataset, where we have evaluated the performance on both retrieving physical cloud properties (liquid water content and liquid water path) and geometric properties (cloud occupancy, cloud base heights, cloud top heights).
>
> As part of the rebuttal, we have additionally quantitatively investigated our model's error across different conditions in our real-world dataset, including different numbers of camera views, cloud coverages, cloud base heights, and cloud geometric thicknesses.
>
> Our key takeaways from this investigation are the following:
>
> (1) The CBH and CTH error increases as the number of views decreases, but the LWC and the LWP are less affected. The decreasing CBH and CTH accuracy with fewer views matches established results in 3D vision, where increasing the number of views improves accuracy in 3D shape estimation.
>
> (2) We observed slightly worse performance for higher cloud base heights and also geometrically thicker clouds, but not to a significant extent.
>
> For additional details, we have included the full numerical results in our response to reviewer XcpS.
>
> &nbsp;
>
> ### Alternative Approaches
>
> >Q10. Have alternative methods such as NERF or DUST3R been considered for 3D estimation? How do these methods compare to the proposed approach?
>
> DUSt3R only reconstructs surfaces, so it is not able to estimate cloud properties across the 3D volume. However, we have tried volumetric methods including MVSNeRF [5], IBRNet [6], and Gaussian Splatting [7]. We found that all of these models severely failed due to the limited number of camera views and because the domain is very different from that on which they were trained.
>
> For MVSNeRF and IBRNet, we additionally tried fine-tuning on our synthetic data but found that this did not significantly improve results. This is not unexpected as these models are generally trained on synthetic datasets with much more than six camera views.
>
> &nbsp;
>
> [1] Large-eddy simulation of the diurnal cycle of shallow cumulus convection over land, Brown et al. (Quarterly Journal of the Royal Meteorological Society 2002)
>
> [2] An efficient parameterization for surface shortwave 3d radiative effects in large-eddy simulations of shallow cumulus clouds, Tijhuis et al. (Journal of Advances in Modeling Earth Systems 2023)
>
> [3] Controls on precipitation and cloudiness in simulations of trade-wind cumulus as observed during rico, vanZanten et al. (Journal of Advances in Modeling Earth Systems 2011)
>
> [4] A large eddy simulation intercomparison study of shallow cumulus convection, Siebesma et al. (Journal of the Atmospheric Sciences 2003)
>
> [5] Mvsnerf: Fast generalizable radiance field reconstruction from multi-view stereo, Chen et al. (ICCV 2021)
>
> [6] IBRNet: Learning Multi-View Image-Based Rendering, Wang et al. (CVPR 2021)
>
> [7] 3D Gaussian Splatting for Real-Time Radiance Field Rendering, Bernhard et al. (SIGGRAPH 2023)

---

> > ### Comment · Reviewer_oz8q · 2025-08-05
> >
> > I had read the reviews from other reviewers and the rebuttals from the authors. I decided to keep my initial assessment and rating of the paper.

---

### Official Review · Reviewer_e4kC · 2025-07-01

**Clarity:** 3
**Significance:** 2
**Originality:** 3
**Rating:** 5
**Confidence:** 2

**Summary:**

This paper presents Cloud4D: a model which predicts cumulus cloud presence, liquid water content, liquid water path, cloud base height and cloud top height from ground based cameras at fine temporal and spatial resolutions. The motivation for this work is to tackle limitations in how such cloud systems are resolved in existing numerical / ML climate models or satellite-based detection models. The Cloud4D modelling framework uses homography and convolutional layers to process image features extracted from ground based camera images by DinoV2 into 2D measures of cloud properties. These cloud properties are then used as initial features in a 3D refinement network that estimates the liquid water content in a voxel. Wind speed is estimated by tracking the movement of reconstructed clouds. The model was trained using a synthetic dataset produced using large eddy simulations and evaluated on a real world dataset captured by cameras and radar over a two month period. The model was shown capture features in ground based observations for liquid water content and path relative to ERA5 reanalysis data and to align with satellite observations of clouds.

**Questions:**

The comparison with the VIP-CT method could have been described and explained in more detail. This setup for using VIP-CT was not introduced in the methods section and, given that VIP-CT is designed for satellite images, how effective is this as a comparison model when applied to ground based camera systems?

It appears as though this system in part relies upon ground based camera systems. Are these just radar images does the model require optical images too? If the model requires optical images, does this mean the system is only capable of operating during daytime? What impact does this have on being able to fully capture the dynamics of cloud systems and to utilise model outputs in downstream tasks?

A related question is how scalable this method is if it relies on ground camera systems? I assume that means it cannot realistically have global coverage? What implications does that have for using the outputs of the Cloud4D model? I would be interested in the authors thoughts and comments on the benefits of investing efforts in furthering ground based systems as opposed to improving satellite-derived monitoring? Is there a middle ground where select ground based systems are used to generate training / evaluation data to refine satellite-based models?

**Ethical Concerns:**

["NO or VERY MINOR ethics concerns only"]

**Final Justification:**

I thank the author's for their engagement and discussion. I have revised my ranking upwards to accept. This is mainly a reflection of the novelty of the approach, the author's explanation of training synthetic data supporting generalisation, and providing additional validation on another dataset (while noting the challenges of accessing validation datasets for this task). I would still flag that the narrow validation undertaken here is a limitation and I felt at times the author's attempts to contextualise the application of this approach was quite general (i.e. suggesting ways it could work at many scales and for many applications as opposed to focusing on the core problem area of relevance).

**Limitations:**

There is a limitations section. However, see the comments in the questions section above regarding discussion of some limitations of this work. My rating now sits between borderline accept and accept. I would err towards the former as, while I found the method to be novel and the author's have explained its relevance / application, I still believe the validation in the paper to be quite limited.

**Paper Formatting Concerns:**

No.

**Quality:**

2

**Strengths And Weaknesses:**

This paper picks off a challenging topic in trying to model cumulus clouds and fine spatial and temporal resolutions. It presents a relatively complex workflow to address this challenge with a model that converts several ground based images, captured at a point in time, into a 3D dataset of several cumulus cloud properties. In the main, this workflow was well presented and easy to follow. The authors also collect a detailed, but localised, ground truth dataset consisting of two months of real world camera observations to evaluate the model. This dataset is a valuable contribution and allows for evaluating the model in considerable detail. However, the localised nature of this dataset left questions as to how generalisable this approach is. From what I could see in the paper or supplementary material, the geographic location or season of this dataset was not presented.

It was good that the authors mentioned limitations of using ground based cameras due to weather-based occlusion. However, there was scope to discuss other limits of using ground based cameras including nighttime captures and scalability and what this means for broad applicability of the Cloud4D approach.

The discussion of the results was quite superficial in places. For example, a figure comparing Cloud4D wind speeds, ERA5 wind speeds and ground truth wind speeds was presented and not discussed. The main results table presents average errors across time and position of clouds / water content / wind speed within the viewed area. It would be good to comment on how much variability there was in model error and the characteristics of the error.

---

> ### Author Rebuttal · Authors · 2025-07-31
>
> We thank the reviewer for their feedback and appreciate that they acknowledge the difficulty of the target task and that they find our dataset to be a valuable contribution. Please find below our response to your comments.
>
> &nbsp;
>
> >Q1. The comparison with the VIP-CT method could have been described and explained in more detail
>
> VIP-CT takes as input satellite-based images ($\mathbb{R}^{1 \times H \times W}$) and outputs a 3D field ($\mathbb{R}^{N_x \times N_y \times N_z}$). This is the same as our setup, with the only difference being that satellite imagery is grayscale while ours is RGB. This only requires a simple change in the number of channels for the first convolutional layer.
>
> We use the publicly available training code for VIP-CT and instead train using our synthetic dataset, which consists of ground-based images rather than satellite-based images.
>
> We will add this description to the paper. Thank you for pointing this out.
>
> &nbsp;
>
> >Q2. VIP-CT is designed for satellite images, how effective is this as a comparison model when applied to ground based camera systems?
>
> A key difference between satellite images and ground-based images is that the viewing angles greatly differ across ground-based views. The lack of explicit camera geometry modeling in VIP-CT can therefore limit its effectiveness with our ground-based images. However, as there is no prior work that predicts volumetric 3D cloud properties from ground-based cameras, we use VIP-CT as the closest comparable model.
>
> &nbsp;
>
> >Q3. Are these just radar images does the model require optical images too?
>
> To clarify, our model estimates full 3D physical cloud properties from **only optical (RGB) images**. No radar images are required, as we only use radar for the evaluation of our model.
>
> &nbsp;
>
> >Q4. If the model requires optical images, does this mean the system is only capable of operating during daytime?
>
> The cameras we have deployed only operate during the daytime. However, we argue this is not a significant limitation as:
>
> (1) Cumulus clouds generally require solar heat to form over land and are therefore mostly present during daytime.
>
> (2) There are alternative cameras that can operate at night and are capable of observing clouds.
>
> &nbsp;
>
> >Q5. How scalable this method is if it relies on ground camera systems? I assume that means it cannot realistically have global coverage?
>
> Our setup is scalable in the sense that it uses commodity cameras, which are cheap, so that new sites can be easily deployed. However, we do not intend for the setup to be deployed in a manner such that global areas are densely monitored by ground-based cameras. We envision them being deployed sparsely at sites of interest to cover large areas. This is because satellites are still the best option for global coverage, and the measurements from Cloud4D are complementary, providing high spatial and temporal resolution measurements for focused areas.
>
> &nbsp;
>
> >Q6. Benefits of investing efforts in furthering ground based systems as opposed to improving satellite-derived monitoring?
>
> We see the ground-based system as complementary to satellites for two reasons. Firstly, the ground-based system has a much higher temporal and spatial resolution. Even with advanced satellites, it would be impossible to get data at 5s timesteps at 25m resolution, useful for fast forming convective clouds like cumulus. Secondly, in the visible spectrum, satellites observe mainly the tops of the clouds, while ground-based cameras are able to provide complementary observations of cloud bases.
>
> &nbsp;
>
> >Q7. Is there a middle ground where select ground based systems are used to generate training / evaluation data to refine satellite-based models?
>
> Absolutely, we believe this is a key potential application of our method. Ground-based cameras can provide additional spatial detail and temporal context, while satellites have the advantage of global spatial coverage.
>
> &nbsp;
>
> >Weakness 1. However, the localised nature of this dataset left questions as to how generalisable this approach is.
>
> We agree that testing the generalization to other sites would be useful. However, we would like to point out that our current model is trained exclusively on synthetic data and is never trained on data from our real-world test site. Only the camera poses in our synthetic data match our real-world setup; the other characteristics are decoupled from the real test site.  Our synthetic dataset includes a wide variety of simulated cloud morphologies, atmospheric conditions, and sun positions that are not unique to our physical location. Therefore, the model is forced to learn the underlying principles of 3D cloud reconstruction from multi-view imagery, rather than overfitting to the specific visual features (e.g., background, typical weather patterns) of a single site. Our real-world site, therefore, acts as a generalization test in itself.
>
> &nbsp;
>
> >Weakness 2. From what I could see in the paper or supplementary material, the geographic location or season of this dataset was not presented.
>
> We have not included the geographic location to minimize identifying information in the double-blind review process. However, we will include precise coordinates and timestamps together with the public release of our real-world dataset.
>
> &nbsp;
>
> >Weakness 3. It would be good to comment on how much variability there was in model error and the characteristics of the error
>
> For the rebuttal, we have investigated our model's error across different conditions in our real-world dataset, including different numbers of camera views, cloud coverages, cloud base heights, and cloud geometric thicknesses.
>
> Our key takeaways from this investigation are the following:
>
> (1) The estimation of CBH and CTH worsens as the number of views decreases, but the LWC and the LWP are less affected. The decreasing CBH and CTH accuracy with fewer views matches established results in 3D vision, where increasing the number of views improves accuracy in 3D shape estimation.
>
> (2) We observed slightly worse performance for higher cloud base heights and also geometrically thicker clouds, but not to a significant extent.
>
> Please see our response to reviewer XcpS for the full numerical results.

---

> > ### Comment · Reviewer_e4kC · 2025-08-05
> >
> > I thank the authors for their very detailed responses. I welcome the additional detail provided on the use of the VIP-CT model and the discussion regarding scalability and the application-space for using ground based cameras. I think this is important detail to make clear in a revised submission. I have remaining queries regarding the validation of the model, which, in turn make it hard to fully appraise its contribution. While the model has been trained on diverse synthetic dataset, it has only been validated at one location. This is a very limited evaluation. While I recognise the logistics and costs associated in setting up ground based camera systems, are there camera datasets which could be used as additional validation data (either public or private)? Do the authors have a strategy for scaling up the validation of the model? Relatedly, I think there is scope to provide more interpretation of the results. While this model performs better than the VIP-CT benchmark, what do the MSE values for the key outcome variables mean in terms of using model outputs for downstream analyses? Are they fit for purpose? I was also curious about Fig 3a - it seems that the model misses some spikes in reference 1D liquid water content. Have the authors explored why?

---

> ### Author Response · Authors · 2025-08-07
>
> We thank the reviewer for their thorough response. Please find below our response to your questions.
>
> &nbsp;
>
> >While I recognise the logistics and costs associated in setting up ground based camera systems, are there camera datasets which could be used as additional validation data (either public or private)?
>
> To the best of our knowledge, SGP-ARM [1] is the only other camera dataset suitable as additional validation data. Below we show results validating our model and VIP-CT on a one-hour segment of cumulus clouds captured in the SGP-ARM dataset:
>
> | &nbsp; &nbsp; &nbsp;  | &nbsp; &nbsp; Occ &nbsp; &nbsp; | &nbsp; LWC (gm⁻³) &nbsp; | &nbsp; LWP (kgm⁻³) &nbsp; | &nbsp; &nbsp; CBH (m) &nbsp; &nbsp; | &nbsp; &nbsp; CTH (m) &nbsp; &nbsp; |
> | :--- | :---: | :---: | :---: | :---: | :---: |
> | VIP-CT &nbsp; &nbsp; | 0.15 | 8.503 | 28.907 | 2257.44 | 1969.44 |
> | Ours &nbsp; &nbsp; | **0.64** | **0.019** | **0.037** | **235.74** | **346.82** |
>
> &nbsp;
>
> Our model shows similar accuracy on this dataset compared to the results on our real-world dataset. We note that VIP-CT performs significantly worse on this dataset, which can be attributed to the fact that it implicitly learns camera geometry and does not generalize to unseen camera poses. On the other hand, our method has no problem transferring to the SGP-ARM camera poses. The generalization across camera setups is due to the homography projection, which helps decouple the cloud estimation task from the camera poses as the learned module operates in world-space. This shows our model’s capability of generalizing to other sites and camera setups.
>
> &nbsp;
>
> >Do the authors have a strategy for scaling up the validation of the model?
>
> The main effort in scaling up the validation would be capturing new data with suitable ground truth. We therefore believe that scaling up the validation is best done through two approaches:
>
> (1) Single-site deployment over time: by deploying the cameras for longer periods, the validation scale will increase in both data quantity and cloud diversity (from seasonal weather changes).
>
> (2) Deployment across multiple sites: this would increase the diversity of observed clouds through geographic changes.
>
> &nbsp;
>
> >What do the MSE values for the key outcome variables mean in terms of using model outputs for downstream analyses? Are they fit for purpose?
>
> To put the error metrics into perspective, typical measurements taken with aircraft probes (which are the standard for in-situ measurements at multiple points throughout a cloud) have an uncertainty of around 10% - 15%. In comparison, our LWC relative error is around a similar range:
>
> $$\dfrac{\text{MAE(LWC of clouds)}}{\text{mean(LWC of clouds from radar GT)}} = \dfrac{0.029 gm^{-3}}{0.321 gm^{-3}} = 8.9\\%$$
>
> We would also like to highlight that there are no existing methods able to estimate dense 3D cloud fields at the spatial and temporal resolution of our method. Our method, therefore, fills an observational gap that is scientifically useful for downstream analyses.
>
> &nbsp;
>
> >I was also curious about Fig 3a - it seems that the model misses some spikes in reference 1D liquid water content. Have the authors explored why?
>
> We believe that the main problem for those spikes is due to there being two cloud layers, where most clouds are at ~1000m, but with a small fraction at ~1750m. As discussed as part of our limitations (Section 7), our model is only trained for single cloud layers and thus misses the second cloud layer.
>
> &nbsp;
>
> [1] Observing Clouds in 4D with Multiview Stereophotogrammetry, David M. Romps and Ruşen Öktem (BAMS 2018)

---

### Official Review · Reviewer_nv5b · 2025-07-02

**Clarity:** 3
**Significance:** 3
**Originality:** 3
**Rating:** 5
**Confidence:** 3

**Summary:**

Authors propose Cloud4D, a method to estimate cloud properties, such as liquid water content (LWC), liquid water path (LWP), cloud base height (CBH) and cloud top height (CTH), from multiple ground-based images.

The method can be summarized as follows:

 * collect N ground-based images of clouds, together with the camera poses and intrinsics;
 * estimate an homography that maps the image space to the cloud plane, in order to estimate the height of the plane the clouds reside on;
 * to estimate cloud properties, DINOv2 is applied to the images, followed by the LoftUp method to reproject the DINOv2 features to the image space. The previously learnt homography projects the features to the cloud plane;
 * the clouds of interest, i.e. cumulus clouds, are characterized by their vertically thin nature, meaning that the main variation in cloud structure happens in the horizontal plane. With this assumption, authors perform the cloud estimation layer as a 2Dto-2D task, using 2D convolutions, which estimate the desired cloud properties;
 * a 3D refinement phase is applied to obtain a 3D estimate from the 2.5D features using a sparse transformer initialized with a basic model of a cloud volume, which is further fused with the projected features derived from DINOv2 and LoftUp, reconstructing 3D profiles for LWC;
 * as a final step, wind motion estimation is provided leveraging the reconstructed 3D LWC cloud profiles and the high temporal resolution (5 s) of the camera. Wind tracking is performed using an off-the-shelf point tracker.

Due to the lack existing datasets on cloud properties, authors validated the method on simulated and observed data, acquired over a two-month deployment of the ground-based cameras. Results show superiority in estimation of cloud properties, although the compared methods are very different in nature and very scarce.

**Questions:**

Here are some questions and suggestions I have for the authors:

 * could the authors provide more context about how the system is meant to be deployed? From the paper is unclear to me whether the system is designed to provide monitoring over large areas, or whether it is rather designed to characterize cumulus clouds at a smaller scale, with the aim of informing the evolution of weather patterns. If the deployment intention is the former, I would like to know about the requirements in terms of number of cameras required to cover, say, a 100kmx100km area, and which communication/collection infrastructure would be required;
 * as mentioned in the weaknesses above, and as the authors recognize in the Limitations Section, I don;t see how the proposed method could be generalized to estimate cloud properties of different cloud types, namely thicker clouds higher in the atmosphere. Such clouds would violate the assumptions and design choices of Cloud4D;
* authors are probably aware, but there are more suited satellite images available than Sentinel-2, e.g. PlanetScope from Planet Labs, which provide a higher temporal revisit time, and often provide multiple takes within minutes of one another. Such imagery could be helpful in validating the cloud profiles, as well as wind tracking module.

**Ethical Concerns:**

["NO or VERY MINOR ethics concerns only"]

**Final Justification:**

I maintain my initial score, as the authors have satisfactorily responded to my questions, and to the ones of my colleagues. My main concerns regarded the deployment at scale of the proposed approach, as well as generalization to other types of cloud.

Authors provided details about possible deployment scenarios, and some pointers on generalization.

**Limitations:**

yes

**Paper Formatting Concerns:**

No major formatting issues.

**Quality:**

3

**Strengths And Weaknesses:**

Estimating cloud properties with adequate spatial and temporal resolution is a very relevant and important task for accurate modeling of weather and climate variables. The paper therefore proposes a very interesting and alternative approach for accurate estimation of 4D cloud properties, clearly outlining the motivation behind design choices and validating the method on simulated and observed datasets.

The paper is original, very well written and clear. Due to the niche application, available validation datasets are scarce, as well as comparable methods. This leads to the use of ad-hoc simulated datasets and a limited observed dataset. Despite this, the potential of the method is in my opinion demonstrated, since authors seek to evaluate their method against very diverse methods for estimation of cloud properties, namely radar measurement, satellite images, and wind profiles.

The main strength I see is the careful design of the imaging and processing system for the modelling of cumulus clouds, which are indeed challenging to characterize with other methods, e.g. remote sensing. However, on the other hand, I feel that this assumption also represents the main weakness of the method, since I don't believe the 2.5D representation and 2D-to-2D cloud property estimation would be valid for more thick types of clouds and clouds higher in the atmosphere.

---

> ### Author Rebuttal · Authors · 2025-07-31
>
> We thank the reviewer for their feedback and appreciate that they find our approach interesting and the target task to be very relevant and important. Please find below our response to your comments.
>
> &nbsp;
>
> >Q1. Could the authors provide more context about how the system is meant to be deployed?
>
> We envision the following possible deployment scenarios:
>
> (1) Small-scale at a single site: this would allow studying cloud physical properties and processes, such as formation and dynamics, which is not possible with existing instruments. It could allow the development and evaluation of parameterisations for weather and climate models. It could also provide a satellite validation site.
>
> (2) Large-scale with sparse coverage: this would be deployed similarly to existing radar and meteorological instrument sites which span a large area but have non-overlapping coverage. The output of this deployment could feed into NWP models or be used for scientific studies.
>
> (3) Large-scale with dense coverage: while dense coverage is more difficult, here we would aim to cover one weather model grid box (about 25km x 25km) to provide a 3D reconstruction of a portion of the cloud field spanning the full grid-box element. This would be useful for validating and improving weather/climate models.
>
> &nbsp;
>
> >Q2. If the deployment intention is the former, I would like to know about the requirements in terms of number of cameras required to cover, say, a 100kmx100km area, and which communication/collection infrastructure would be required
>
> A dense deployment covering a 25km x 25km area would require the following:
>
> **Number of cameras**. The most basic scaling of our deployment would be to duplicate our current camera setup across a grid. As our current setup is covering 5km x 5km using 6 cameras, scaling up to the sizes of a weather model gridbox ~(25km x 25km) would require 6 * 25 = 150 cameras. Given that our cameras are commodity cameras, this would still be cheaper than existing measurement devices such as radars. For future work, it could be of interest to investigate alternative camera setups for larger areas.
>
> **Infrastructure**. The cameras have low power consumption and are powered by an external portable solar panel. Time synchronization and transfer of camera data to an external server is done using a standard 4G cellular connection through a Raspberry Pi.
>
> &nbsp;
>
> >Q3. I don’t see how the proposed method could be generalized to estimate cloud properties of different cloud types, namely thicker clouds higher in the atmosphere
>
> We would like to highlight that while an *individual cloud* might be more vertically thick for other cloud types, the vertical thin and horizontally wide structure still holds for any overall cloud *field*.
>
> This vertical structure is a property of the atmosphere in general and also motivates why weather models such as the Integrated Forecasting System (ECMWF) and GraphCast [1] have gridboxes which are also vertically thin and horizontally wide.
>
> &nbsp;
>
> >Q4. There are more suited satellite images available than Sentinel-2, e.g. PlanetScope from Planet Labs
>
> PlanetScope is a good candidate for testing in the future, but as data access is only granted on request, we have chosen to compare against MODIS and Sentinel-2, as their data is publicly available and commonly used for cloud physics research.
>
> &nbsp;
>
> [1] Learning skillful medium-range global weather forecasting, Lam et al. (Science 2023)

---

> > ### Comment · Reviewer_nv5b · 2025-08-01
> >
> > I thank the authors for their clarifications. Having read the reviews from reviewers, and the rebuttals from the authors, I keep my initial assessment and rating of the paper.

---

### Official Review · Reviewer_XcpS · 2025-07-02

**Clarity:** 1
**Significance:** 3
**Originality:** 3
**Rating:** 4
**Confidence:** 4

**Summary:**

The paper presents Cloud4D, a novel machine learning framework designed to estimate high-resolution, four-dimensional (3D space + time) cloud properties using synchronized ground-based cameras.
Authors mention a three-fold Contribution:
* Cloud4D estimates liquid water content (LWC), cloud base height (CBH), and cloud top height (CTH) at a 25 m spatial and 5 s temporal resolution—a significant improvement over satellite and radar methods.
* The system uses a transformer model guided by camera homographies to map 2D images into structured 3D cloud layer representations.
*  A demonstration, on a two-month deployment with six cameras, that Cloud4D delivers an order-of-magnitude improvement in temporal resolution over space-borne products while achieving < 10% relative error against collocated radar retrievals.

The method to achieve the 3D LWC comprises of two main steps:
* A 2.5D cloud layer model for estimating cloud layer parameters.
* A sparse 3D transformer that refines these into 3D predictions.

Finally an Horizontal Wind Estimation is presented. By tracking cloud movement over time in the 3D reconstruction, Cloud4D infers height-resolved horizontal wind vectors.

**Questions:**

* Important: Justification of 10% Relative Error - The abstract and main text claim "<10% relative error against collocated radar retrievals," but this metric does not appear in any table or figure. Where is this value calculated, and can you provide quantitative evidence or direct references to support this claim?

* Moderate: It seems that the training of Cloud4D is performed exclusively on synthetic data. Isn’t there a need to address the domain gap between synthetic and real-world observations when deploying the model on real camera and radar data?

* Moderate: Did you consider implementing and comparing with other models (e.g., multi-view stereo, which is also referenced in your work)?

* Minor: Ambiguity of "K" Symbol: In Section 3.1 and related equations, does the symbol "K" refer to the number of height planes sampled, the number of camera intrinsics, or both? Could you clarify its precise meaning and usage in the model pipeline?

**Ethical Concerns:**

["NO or VERY MINOR ethics concerns only"]

**Final Justification:**

As concluded in my comment to authors rebuttal and considering the latest comments of the other reviewers I'm raising my rating.

**Limitations:**

While Cloud4D achieves much higher spatial and temporal resolution than satellites, it requires dense, site-specific camera deployments. In contrast, satellite data, though coarser, can be applied globally and generalized to any location without extra infrastructure. This makes Cloud4D less scalable and less universally applicable than satellite-based approaches.

**Quality:**

2

**Strengths And Weaknesses:**

Strengths
* The proposed methods achieves 25 m spatial and 5 s temporal resolution, an order-of-magnitude better than satellite products.
* The authors introduce a homography-guided 2D-to-3D transformer pipeline, leveraging DINOv2, LoftUp, and a sparse transformer for efficient and accurate 3D cloud field estimation.
* Height-resolved horizontal wind profiles are derived by tracking reconstructed 3D cloud fields over time, a capability not available in previous multi-view or satellite-based methods.
* Demonstration of the method on a two-month deployment with six cameras, showing practical feasibility.
* Using synthetic data from MicroHH (atmospheric simulation) and Terragen (cloud rendering) lets Cloud4D train and test its models on realistic, fully labeled 3D cloud scenes that aren’t possible to capture in the real world.

Main Weaknesses:
* The abstract and introduction repeatedly claim <10% relative error against collocated radar measurements, but this number does not appear in any table, figure, or explicit result in the main or supplementary material.  There are no clear, quantitative error tables or plots showing this metric, making it difficult to verify the headline claim.
* The paper lacks comprehensive quantitative comparisons (e.g., with other methods or baselines) and detailed error analysis across different cloud types, conditions, or time periods.
*  Generalizability to other sites, camera setups, number of views or cloud regimes is not tested or discussed in depth. Of course, this is an inherent limitation of the method, since deploying many sites and setups requires significant effort.
* The method relies on radar retrievals for ground truth, which themselves have limitations and may not always be available or accurate, especially for shallow clouds.

Conclusion
* This paper presents an innovative and state-of-the-art approach for 3D cloud reconstruction, leveraging a sophisticated 2D-to-3D transformer architecture and foundation models to achieve unprecedented spatial and temporal resolution. The use of only synthetic data for training the model  (even though it can raise domain adaptation concerns) is particularly interesting and, based on the partial comparisons to radar retrievals shown in Figure 3 and the quantitative results in Table 1, appears promising.
* The main concern is the lack of direct evidence for the headline "<10% relative error" claim. This is repeated in the abstract and introduction, but to my understanding, no table, figure, or quantitative analysis substantiates it. I recommend a borderline reject, but encourage the authors to address this, as the work has significant potential.

---

> ### Author Rebuttal · Authors · 2025-07-31
>
> We thank the reviewer for their feedback, and we appreciate that they find our approach to be innovative and with significant potential. Please find below our response to your comments.
>
> &nbsp;
>
> >Q1. Important: Justification of 10% Relative Error
>
> Thank you for pointing this out. We agree that it is important for this to be clear.
> The relative error specifically refers to our LWC predictions when compared to radar values and is calculated as
>
> $$\dfrac{\text{MAE(LWC of clouds)}}{\text{mean(LWC of clouds from radar GT)}} = \dfrac{0.029 gm^{-3}}{0.321 gm^{-3}} = 8.9\\%$$
>
> where the mean absolute error of our LWC predictions is the same as the value from Table 1 in the paper.
>
> We will make this clearer in the paper.
>
> &nbsp;
>
> >Q2. Moderate: Isn’t there a need to address the domain gap between synthetic and real-world observations when deploying the model on real camera and radar data?
>
> It would be ideal to train our model with real-world data, but there are currently no observing systems that are able to provide the full 3D observations of cloud fields at the spatial scale required. To fill this gap, we use high-resolution LES, which is a state-of-the-art simulation approach for clouds. To ensure that our model generalizes across different conditions, we simulate a number of cases with different meteorological environments.
>
> Our results validate the effectiveness of LES-based training, showing that real-world data is not required, although it may improve results further. For future work, it could be interesting to investigate how real-world data might be integrated with simulation-based training.
>
> &nbsp;
>
> >Q3. Moderate: Did you consider implementing and comparing with other models (e.g., multi-view stereo, which is also referenced in your work)?
>
> Multi-view stereo only reconstructs surfaces, so it is not able to estimate cloud properties across the 3D volume. However, we have tried volumetric methods including MVSNeRF [1], IBRNet [2], and Gaussian Splatting [3]. We found that all of these models severely failed due to the limited number of camera views and because the domain is very different from that on which they were trained.
>
> For MVSNeRF and IBRNet, we additionally tried fine-tuning on our synthetic data but found that this did not significantly improve results. This is not unexpected as these models are generally trained on synthetic datasets with much more than six camera views.
>
> &nbsp;
>
> >Q4. Minor: In Section 3.1 and related equations, does the symbol "K" refer to the number of height planes sampled, the number of camera intrinsics, or both?
>
> Thank you for pointing this out. K refers to the number of height planes sampled, and we will update the notation to avoid confusion with the camera intrinsics that are denoted with a bolded K. We denote the number of images (which are the same as the number of intrinsics) as N.
>
> &nbsp;
>
> >Weakness 1. Generalizability to other sites, camera setups, number of views or cloud regimes is not tested or discussed in depth. Of course, this is an inherent limitation of the method, since deploying many sites and setups requires significant effort.
>
> We agree that testing the generalization to other sites would be useful. However, we would like to point out that our current model is trained exclusively on synthetic data and is never trained on data from our real-world test site. Only the camera poses in our synthetic data match our real-world setup; the other characteristics are decoupled from the real test site.  Our synthetic dataset includes a wide variety of simulated cloud morphologies, atmospheric conditions, and sun positions that are not unique to our physical location. Therefore, the model is forced to learn the underlying principles of 3D cloud reconstruction from multi-view imagery, rather than overfitting to the specific visual features (e.g., background, typical weather patterns) of a single site. Our real-world site, therefore, acts as a generalization test in itself.
>
> We have also investigated our model's performance across different conditions in our real-world dataset, including different numbers of camera views, cloud coverages, cloud base heights, and cloud geometric thicknesses. These results are included below, where camera pairs are randomly and uniformly dropped, and the radar GT has been used to classify the cloud coverage, mean cloud base height, and mean cloud geometric thickness, over one-hour segments in our real-world dataset.
>
> Our key takeaways from this investigation are the following:
>
> (1) The estimation of CBH and CTH worsens as the number of views decreases, but the LWC and the LWP are less affected. The decreasing CBH and CTH accuracy with dropped views matches established results in 3D vision, where increasing the number of views improves accuracy in 3D shape estimation.
>
> (2) We observed slightly worse performance for higher cloud base heights and also geometrically thicker clouds, but not to a significant extent.
>
> | Number of Dropped Views &nbsp; &nbsp; &nbsp; | &nbsp; &nbsp; Occ &nbsp; &nbsp; | &nbsp; LWC (gm⁻³) &nbsp; | &nbsp; LWP (kgm⁻³) &nbsp; | &nbsp; &nbsp; CBH (m) &nbsp; &nbsp; | &nbsp; &nbsp; CTH (m) &nbsp; &nbsp; |
> | :--- | :---: | :---: | :---: | :---: | :---: |
> | 0 | 0.70 | **0.029** | **0.063** | **189.58** | **295.77** |
> | 2 | **0.76** | 0.038 | 0.070 | 316.28 | 371.71 |
> | 4 | 0.68 | 0.037 | 0.072 | 325.03 | 383.16 |
>
> &nbsp;
>
> | Cloud Coverage &nbsp; &nbsp; &nbsp; &nbsp; &nbsp; &nbsp; | &nbsp; &nbsp; Occ &nbsp; &nbsp; | &nbsp; LWC (gm⁻³) &nbsp; | &nbsp; LWP (kgm⁻³) &nbsp; | &nbsp; &nbsp; CBH (m) &nbsp; &nbsp; | &nbsp; &nbsp; CTH (m) &nbsp; &nbsp; |
> | :--- | :---: | :---: | :---: | :---: | :---: |
> | 0.20 - 0.45 | 0.65 | 0.034 | 0.077 | **149.36** | **287.12** |
> | 0.75 - 0.90 | **0.75** | **0.023** | **0.048** | 230.92 | 304.67 |
>
> &nbsp;
>
> | Mean CBH (m) &nbsp; &nbsp; &nbsp; &nbsp; &nbsp; &nbsp; &nbsp; &nbsp; | &nbsp; &nbsp; Occ &nbsp; &nbsp; | &nbsp; LWC (gm⁻³) &nbsp; | &nbsp; LWP (kgm⁻³) &nbsp; | &nbsp; &nbsp; CBH (m) &nbsp; &nbsp; | &nbsp; &nbsp; CTH (m) &nbsp; &nbsp; |
> | :--- | :---: | :---: | :---: | :---: | :---: |
> | 800 - 1350 | **0.74** | **0.027** | **0.055** | 198.63 | **281.29** |
> | 1350 - 1650 | 0.63 | 0.033 | 0.077 | **173.36** | 321.73 |
>
> &nbsp;
>
> | Mean Cloud Thickness (m) &nbsp; &nbsp; &nbsp; | &nbsp; &nbsp; Occ &nbsp; &nbsp; | &nbsp; LWC (gm⁻³) &nbsp; | &nbsp; LWP (kgm⁻³) &nbsp; | &nbsp; &nbsp; CBH (m) &nbsp; &nbsp; | &nbsp; &nbsp; CTH (m) &nbsp; &nbsp; |
> | :--- | :---: | :---: | :---: | :---: | :---: |
> | 150 - 275 | 0.69 | **0.025** | **0.050** | 194.84 | **269.58** |
> | 275 - 500 | **0.71** | 0.033 | 0.076 | **184.09** | 323.12 |
>
> &nbsp;
>
> >Weakness 2. The method relies on radar retrievals for ground truth, which themselves have limitations and may not always be available or accurate, especially for shallow clouds.
>
> We agree that radars have their own biases and limitations. We would like to note that the liquid water content retrieval of the radar also includes information from a coincident lidar and is also scaled to match a microwave radiometer [4]. It is the best possible evaluation dataset for this deployment, but we note the possibility of errors in this data.
>
> &nbsp;
>
> [1] Mvsnerf: Fast generalizable radiance field reconstruction from multi-view stereo, Chen et al. (ICCV 2021)
>
> [2] IBRNet: Learning Multi-View Image-Based Rendering, Wang et al. (CVPR 2021)
>
> [3] 3D Gaussian Splatting for Real-Time Radiance Field Rendering, Bernhard et al. (SIGGRAPH 2023)
>
> [4] Cloudnet: Continuous Evaluation of Cloud Profiles in Seven Operational Models Using Ground-Based Observations, Illingworth et al. (AMS 2007)

---

> > ### Comment · Reviewer_XcpS · 2025-08-06
> >
> > This paper addresses the reconstruction of clouds from multiview cameras, successfully representing the liquid water content in 3D and the wind vectors (the 4th dimension) at unique spatial and temporal resolutions of 25 m and 5 seconds, respectively. The authors present a well-designed deep learning architecture that leverages the capabilities of DINOv2, with the necessary modifications tailored to the specific problem.
> >
> > The authors have satisfactorily addressed my most important concerns:
> > 1. Authors clarified the calculation of the <10% Relative Error, supporting a central claim of the paper. Furthermore they provide to reviewer fVbw the mean and 2 sigma error over 5 (shorter in steps) runs that demonstrate the robustness of the results.
> > 2.  Authors justified the exclusive use of synthetic data for training, as it remains the best available option to date.
> > 3.  Authors answered to me about other models tested that produced poor results, not worth including in the paper, and to reviewer e4kC about the limitations of VIP-CT, which is presented in the paper, because of the different viewing angles between ground cameras and the satellites.
> >
> > The assumptions made for using synthetic data in training and radar data as ground truth for testing are reasonable, given the lack of better proven alternatives. Moreover, in their discussion with the other reviewers, the authors acknowledge the limitations of their method for daytime-only operation and the challenges of global implementation. However, they note that deployment at a regional scale (around 100 km) is feasible and that expanding the method to include cloud types beyond shallow clouds is part of their ongoing research roadmap.
> >
> > While there are clear limitations in applying this method on a global scale, I believe local implementations could significantly enhance our understanding of cloud formation and movement, serving as a step toward more accurate weather and precipitation prediction.
> >
> > For these reasons, I am inclined to increase my rating.

---

### Decision · Program_Chairs · 2025-09-17

**Decision:**

Accept (spotlight)

**Comment:**

Summary: Cloud4D reconstructs 4D cumulus cloud fields + winds from six synchronized ground cameras.  The resolutiion: 25 m spatial and 5 s temporal resolution is better than alternative approaches.

Strengths:
Clear, novel geometry-aware pipeline (DINOv2+LoftUp+homography gives sparse 3D refinement)
Order-of-magnitude space-time resolution gain over satellites.
Good real world demo.

Weaknesses:
Real-world validation is narrow (single primary site, daytime, mostly shallow single-layer cumulus).
Limited baselines (VIP-CT adapted; NeRF-style methods fail but comparisons remain thin).

Decision and justification:
Substantial, original capability: high-resolution, physically meaningful 4D cloud reconstructions from commodity cameras.
Method is technically sound.